# KVQuant: Towards 10 Million Context Length LLM Inference with KV Cache Quantization

Coleman Hooper[1]    Sehoon Kim[1]    Hiva Mohammadzadeh[1]
Michael W. Mahoney[1,2,3]    Yakun Sophia Shao[1]    Kurt Keutzer[1]    Amir Gholami[1,2]

[1]University of California, Berkeley   [2]ICSI   [3]LBNL
{chooper, sehoonkim, hiva, mahoneymw, ysshao, keutzer, amirgh}@berkeley.edu

## Abstract

LLMs are seeing growing use for applications which require large context windows, and with these large context windows KV cache activations surface as the dominant contributor to memory consumption during inference. Quantization is a promising approach for compressing KV cache activations; however, existing solutions fail to represent activations accurately in sub-4-bit precision. Our work, KVQuant, facilitates low precision KV cache quantization by incorporating several novel methods: (i) *Per-Channel Key Quantization*, where we adjust the dimension along which we quantize the Key activations to better match the distribution; (ii) *Pre-RoPE Key Quantization*, where we quantize Key activations before the rotary positional embedding to mitigate its impact on quantization; (iii) *Non-Uniform KV Cache Quantization*, where we derive per-layer sensitivity-weighted non-uniform datatypes that better represent the distributions; and (iv) *Per-Vector Dense-and-Sparse Quantization*, where we isolate outliers separately for each vector to minimize skews in quantization ranges. By applying our method to the LLaMA, Llama-2, Llama-3, and Mistral models, we achieve $< 0.1$ perplexity degradation with 3-bit quantization on both Wikitext-2 and C4, outperforming existing approaches. Our method enables serving LLaMA-7B with a context length of up to **1 million on a single A100-80GB GPU** and up to **10 million on an 8-GPU system**. We develop custom CUDA kernels for KVQuant, showing that we can achieve up to $\sim 1.7\times$ speedups, compared to baseline fp16 matrix-vector multiplications, for the LLaMA-7B model. Code is available at https://github.com/SqueezeAILab/KVQuant.

## 1   Introduction

Large language models (LLMs) have revolutionized many natural language processing (NLP) tasks. In order to improve the capabilities of LLMs, there is significant interest in increasing the context lengths of LLMs. Longer context lengths enable new applications, including long document summarization, retrieval for answering questions about long documents, extended multi-turn applications [6], and code analysis. To support this pull from applications, there have been significant recent advances in long-context length models in industry [2, 30], as well as in academia [6, 22].

Given the importance of LLM workloads, there is strong motivation to improve their inference efficiency. LLM inference with large context lengths can be incredibly resource-intensive; serving LLMs requires high-end GPUs, and the largest LLMs require costly multi-GPU inference setups. When analyzing the computational nature of generative inference with LLMs, it becomes quickly apparent that, for relatively small batch sizes, the computation is memory bound [17]. With the growing divergence between computational speeds and memory speeds, this problem is only going to get worse over time [12]. This makes reducing the memory bottleneck preeminently important. Further analysis shows that the memory bottleneck is strongly related to context size. For short

38th Conference on Neural Information Processing Systems (NeurIPS 2024).

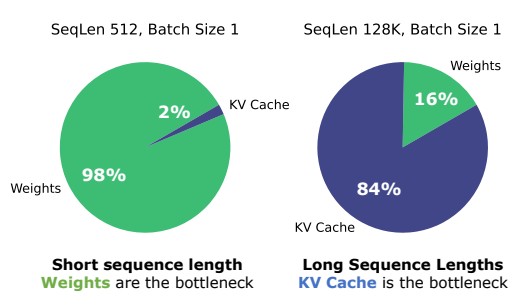 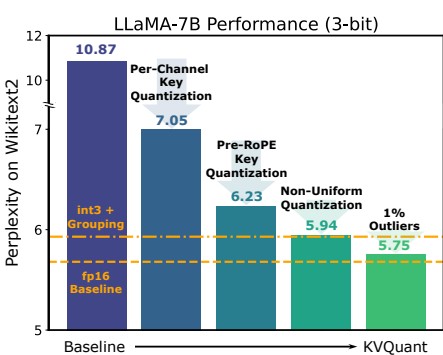

**Figure 1:** *Left: Model size versus activation memory size for the LLaMA-7B model with sequence length 512 and 128K. For longer context lengths, the KV cache becomes the dominant memory bottleneck. Memory consumption of model weights and KV cache activations for different LLaMA models with different sequence lengths are provided in Table 7 in Appendix A. Right: Overview of the different components used in KVQuant that result in less than 0.1 perplexity degradation over the fp16 baseline when quantizing the KV cache for the LLaMA-7B model to 3-bit precision. As shown in Table 1, our 3-bit approach results in $4.8\times$ reduction in cached activation memory footprint.*

sequence lengths, the dominant contributor to memory consumption is the weight matrices, and therefore the optimal strategy is to minimize the model size in order to reduce memory consumption as well as bandwidth requirements [18, 17]. However, as shown in Figure 1, the main bottleneck for long sequence lengths is the memory requirements for caching Key and Value (KV) activations throughout inference. This challenge is further exacerbated when one considers batched inference.

It is therefore crucial to develop methods for compressing the KV cache to enable efficient long-sequence length inference. Existing approaches lead to unacceptable accuracy degradation due to the outlier structures in KV cache activations as well as suboptimal bit allocation with existing uniform and non-uniform approaches. In this work, we perform an extensive analysis of KV cache activations in recent LLMs, revealing patterns which can be exploited to enable ultra-low precision quantization with minimal accuracy loss. In particular, we make the following contributions (summarized in Figure 1):

- We find that the Keys exhibit outliers in specific channels before applying RoPE. However, the outlier channel magnitudes become less consistent after applying RoPE, posing a distinct challenge for low precision quantization. We address this by quantizing Keys per-channel before RoPE is applied (see Section 3.1 and Section 3.2).

- We find that existing uniform and non-uniform quantization methods result in sub-optimal quantization signpost placement. Instead, we propose a Non-Uniform Quantization (NUQ) method which considers sensitivity and not just magnitude when quantizing activations. We show that one can apply sensitivity-weighted non-uniform quantization offline on a calibration set to derive accurate datatypes for KV cache quantization (see Section 3.3).

- Even with the above, we find that outlier values in cached KV activations can significantly degrade quantization resolution. Unlike for weights, it is non-trivial to extract outlier values from activations, given the dynamic nature of activations. However, we find that we can efficiently and accurately identify and compress outlier values in order to store them compactly in a separate sparse representation. We also find that per-vector outlier detection outperforms per-matrix outlier detection with no additional memory overhead. By removing only 1% of outliers, we can attain under 0.1 perplexity degradation on both Wikitext-2 and C4 for 3-bit KV cache quantization with the LLaMA, Llama-2, Llama-3, and Mistral models, thereby facilitating accurate inference with $4.8\times$ longer context length.

- We implement custom CUDA kernels to perform activation quantization efficiently during inference, achieving up to $\sim1.7\times$ speedups for Key and Value matrix-vector multiplications for LLaMA-7B at 4-bit precision relative to the fp16 baseline (see Section 3.7 and 4.4). These results demonstrate how our methodology allows for accurate and efficient low-bit KV cache quantization.

## 2 Background

### 2.1 LLM Inference

When inferring a decoder-only LLM, inference proceeds in two distinct phases. In the prefill phase, the model takes in an input prompt, which it processes *in parallel*. During the generation phase, the model then generates the output sequence autoregressively, meaning that each token generation is dependent on all previously generated tokens. As such, for small batch sizes, the generation phase of LLM inference is typically *memory-bandwidth bound*, as the only available parallelism is across different sequences in a given batch. Additionally, during generation, the model needs to store intermediate Key and Value activations at each layer in order to condition generations on previously generated output tokens. Otherwise, we would need to recompute all prior Keys and Values at each timestep, which would be prohibitively expensive. These stored activations are referred to as the *Key-Value (KV) cache*. Throughout this paper, we will capitalize Key and Value to distinguish when we are referring to the KV cache tensors. Assuming a model with $n$ layers and $h$ attention heads with dimension $d$ that is stored using $e$ bytes per element, the KV cache size for batch size $b$ and sequence length $l$ is $2 \cdot n \cdot h \cdot d \cdot e \cdot b \cdot l$, meaning that it grows linearly with both batch size and sequence length. As shown in Table 7, the KV cache becomes the dominant contributor to memory consumption for longer sequence lengths and larger batch sizes. Note that since each sequence in batched inference depends on separate past context, there is no available batch-level parallelism when loading the cached Keys and Values for their respective computations in batched inference. KV cache loading is therefore always *memory-bandwidth bound*. This motivates pursuing methods to optimally compress the KV cache, even at the expense of a more complex dequantization process.

### 2.2 LLM Quantization

There have been many prior works on LLM quantization. Several have focused on weight-only quantization for LLMs, due to the greater contribution to memory consumption and runtime for small sequence lengths and batch sizes [21, 9, 17]. Prior works have noted the presence of distinct outliers in both weights and activations [7, 9, 17]. One approach that has been developed to address this outlier issue in the context of weight quantization is dense-and-sparse quantization, where each weight matrix is decomposed into a sparse outlier matrix and a dense low-precision matrix [9, 17]. Prior works have also leveraged non-uniform quantization methods to improve accuracy for the same bit precision by allowing for flexible quantization signpost placement [17, 8]. These approaches have either used a fixed non-uniform datatype such as NormalFloat [8], or derived quantization signposts using a sensitivity-weighted k-means approach [17]. Appendix B provides a more detailed overview of related work for outlier-aware LLM quantization and non-uniform LLM quantization.

There has also been work on quantizing both weights and activations (including KV cache) [41, 33]. However, there is still a significant perplexity degradation when quantizing KV cache activations to low precision; [34, 44] quantized KV cache activations to 4-bits, but required fine-grained grouping for 4-bit quantization, while still observing some perplexity degradation, and [34] observed that 3-bit KV cache quantization with fine-grained grouping leads to unacceptable accuracy loss. Other works quantized KV cache activations to 4-bits but required retraining to maintain performance [24]. One concurrent work also explores low precision KV cache quantization in order to enable larger batch size inference by reducing the KV cache size [26].

### 2.3 KV Cache Compression

There have also been several prior works on compressing the KV cache. Some of these methods aim to only store important tokens in the KV cache and to evict less important tokens, thereby maintaining low memory usage [25, 43, 11, 20]. Other methods aim to only retrieve a subset of tokens at each step to achieve memory bandwidth savings [32]. In this work, we explore KV cache quantization as an orthogonal direction for compressing the KV cache in order to enable long context inference.

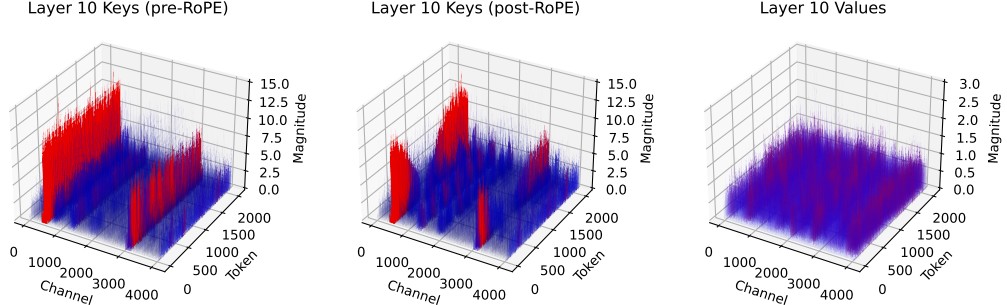

**Figure 2:** *Example distributions of the activation values for Keys pre-RoPE, Keys post-RoPE, and Values for LLaMA-7B on a sample with 2K sequence length from Wikitext-2. We observe several patterns: (i) Keys pre-RoPE exhibit clear outliers in specific channels across different tokens; (ii) after applying RoPE, the distribution becomes less structured and there are less consistent magnitudes for outlier channels (this is expected, as RoPE applies a rotation operation between pairs of channels); and (iii) Values exhibit no fixed outlier pattern with outlier values across channels and tokens.*

## 3 Method

### 3.1 Per-Channel Key Quantization

To inform our approach, we first performed a detailed analysis to understand the KV cache distributions. Figure 2 shows sample distributions for the KV cache activations. We observe that the Key matrices tend to have distinct outlier channels, which have larger average magnitudes than other channels; this corroborates previous observations about outlier channels in LLM activations [7, 41]. The Value matrices exhibit both outlier channels as well as outlier tokens (although these outliers are less extreme than the outlier Key channels).

Existing KV cache quantization approaches perform per-token quantization (meaning that the scaling factor and zero-point are shared by elements in the same token) [34, 44]. However, due to the differing average magnitudes between channels, the values within a channel are easier to quantize when grouped together than the values across different channels. As such, to better match the distributions, we investigate *per-channel* KV cache quantization, meaning that the scaling factor and zero-point are shared by elements in the same channel. By sharing the scaling factor and zero-point along the channel dimension, this will naturally group together values with similar magnitudes, thereby mitigating the impacts of outlier channels on other channels when quantizing to low precision. As outlined in Appendix G, we find that per-channel quantization provides significant accuracy benefits for Keys but not for Values. By leveraging per-channel quantization for Keys and per-token quantization for Values, we observe a 3.82 perplexity improvement on Wikitext-2 for 3-bit LLaMA-7B quantization. Note that this can potentially add runtime overhead since the quantization dimension is now misaligned with the reduction dimension for the Keys when performing matrix-vector multiplications. However, we find that we are able to efficiently dequantize Keys and perform the Query-Key matrix-vector multiplication without adding runtime overhead, as shown in Section 4.4. Additionally, as outlined in Section 3.6, per-channel quantization can also be challenging due to the need to recompute scaling factors as tokens are added to the Key cache. We show that we can calibrate offline for scaling factors, thereby avoiding expensive online recomputation.

Per-channel Key quantization was also explored in another concurrent work [26], which leveraged similar intuition about grouping together large magnitude values in the same channel to minimize quantization error. Their methodology requires fine-grained grouping for per-channel quantization while maintaining a residual subset of the KV cache in fp16. In our work, we demonstrate that by leveraging offline calibration, we can accurately perform per-channel quantization without grouping.

### 3.2 Pre-RoPE Key Quantization

One issue when quantizing Keys is handling the rotary positional embedding (RoPE), which is applied to Keys and Queries in most public LLMs, including LLaMA and Llama-2 [35]. Given Query and Key vectors $Q_m = W_q * x_m$ and $K_n = W_k * x_n$ at positions $m$ and $n$ in the sequence, RoPE is applied as position-dependent rotations to each of these vectors to obtain $\tilde{Q}_m = R^d_{\theta,m} \cdot Q_m$ and

$\tilde{K}_n = R_{\theta,n}^d \cdot K_n$. This embeds the relative position between a Query and Key vector as an amount of an angle that is a multiple of its position index. When caching Key vectors, we therefore need to either cache $\tilde{K}_n$, or else we need to cache $K_n$ and apply $R_{\theta,n}^d$ on-the-fly during inference. The challenge with caching Key vectors after applying this rotation is that it leads to mixing pairs of channels by different amounts for different positions in the sequence, as shown in Appendix C (since it jointly rotates pairs of channels by different angles depending on the position in the sequence). The post-RoPE activation distribution is also shown in Figure 2, demonstrating how the rotation between pairs of channels leads to less consistent channel magnitudes. This makes it harder to quantize Key activation channels which would typically have consistent large-magnitude values. This motivated our investigation into whether we could perform *pre-RoPE* Key quantization (meaning that we quantize $K_n$) and then efficiently apply the positional embeddings on-the-fly after dequantization. The benefits of pre-RoPE Key quantization are highlighted in Appendix H, yielding 0.82 perplexity improvement on Wikitext-2 for 3-bit LLaMA-7B quantization. To be able to quantize Keys pre-RoPE, we develop a fused kernel to efficiently apply RoPE post-dequantization (the details of this approach will be discussed in Section 3.7).

### 3.3 nuqX: An X-Bit Per-Layer Sensitivity-Weighted Non-Uniform Datatype

Uniform quantization is suboptimal for KV cache quantization since the Query and Key activations are non-uniform. Additionally, KV cache loading is memory bandwidth bound, regardless of batch size or sequence length, meaning that the dequantization overhead introduced by non-uniform quantization methods is not problematic (since the added computation does not introduce any additional latency). It is therefore desirable to leverage non-uniform quantization methods for KV cache quantization. In [17], the authors computed non-uniform quantization signposts using a sensitivity-weighted k-means approach. However, this cannot be applied directly to KV cache quantization as the Values are quantized dynamically at runtime, which means that we would need to apply K-means online during inference, and it is also difficult to estimate sensitivity for activations online. We therefore facilitate efficient online non-uniform KV cache quantization by deriving a per-tensor non-uniform datatype offline on calibration data, which is then rescaled per-channel or per-token to accurately represent the key and value distributions. We compute sensitivity-weighted quantization signposts offline on a calibration set prior to inference, while maintaining compatibility with per-vector quantization by separately normalizing each channel to the range $[-1, 1]$ prior to deriving the shared datatype. Using the diagonal Fisher information matrix (derived in Appendix D), along with the quantization error for activation $A$, we formulate the error minimization objective as follows, where $A$ is flattened to one dimension and where $N$ is the number of elements from all of the samples in our calibration set:

$$Q(A)^* \simeq \arg\min_Q \sum_{i=1}^N \mathcal{F}_{ii} \big(A_i - Q(A_i)\big)^2. \tag{1}$$

We modify the objective in Equation 1 as described in Appendix E in order to apply it using the normalized activation values. We then minimize it offline on a calibration set using a k-means solver in order to obtain the quantization signposts for the non-uniform datatype for each Key or Value layer. Appendix I compares our non-uniform quantization approach with existing uniform and non-uniform quantization baselines [8], demonstrating how our non-uniform approach provides 0.29 perplexity improvement on Wikitext-2 for LLaMA-7B relative to 3-bit uniform methods. Table 16 in Appendix L shows how computing the required Fisher information for the LLaMA-65B model takes only a few minutes, and how using the k-means solver takes only a few minutes per layer (with the computation for each layer being parallelizable).

### 3.4 Per-Vector Dense-and-Sparse Quantization

As shown in Figure 4 in Appendix F, for both Keys and Values, the majority of elements are contained within a small percentage of the dynamic range. This means that by leveraging dense-and-sparse quantization, as demonstrated in [17], in order to isolate a small percentage of numerical outliers, we can restrict the range that we need to represent, thereby allowing us to represent the remaining elements with greater precision. However, when looking at the Key and Value distributions in Figure 2, different channels and tokens have different average magnitudes. Therefore, an element which counts as an outlier in one channel may not be an outlier in another channel (since that channel

may have a greater average magnitude), making naive application of dense-and-sparse quantization suboptimal. It is therefore crucial to directly target the outlier values that skew the dynamic range *at the granularity that we are quantizing* in order to address the values that are exaggerating the range along that particular dimension. In this work, we leverage *per-vector* dense-and-sparse quantization, where we use a different outlier threshold per-vector (either a separate threshold per-channel for per-channel quantization, or a separate threshold per-token for per-token quantization), rather than a single outlier threshold for each layer.

Note that computing outlier thresholds for per-vector dense-and-sparse quantization poses potential accuracy and efficiency challenges. However, in Section 3.6, we show that we are able to accurately calibrate for per-channel outlier thresholds offline and efficiently compute per-token outlier thresholds online. After determining the upper and lower outlier thresholds, the remaining numbers in the vector are normalized to the range $[-1, 1]$, and we then minimize Equation 1 (ignoring outliers) in order to obtain the quantization signposts for the non-uniform datatype for the remaining numbers. Appendix J will demonstrate the benefits of removing a small percentage of numerical outliers and keeping them in full precision, as well as the advantages of per-vector dense-and-sparse quantization over using a single global outlier threshold for each layer. As shown in Figure 1, by removing 1% of numerical outliers using per-vector outlier thresholds, we achieve an additional 0.19 perplexity improvement on Wikitext-2 for 3-bit LLaMA-7B quantization, which is within 0.07 perplexity of the fp16 baseline.

### 3.5 Attention Sink-Aware Quantization

Prior work has demonstrated that after the first few layers in LLMs, the model tends to allocate a large attention score to the first token [42]. This occurs even when the initial token is not semantically important. This phenomenon happens because the model tends to use the inital token as a "sink". In our work, we demonstrate that due to the Attention Sink phenomenon, the model is disproportionately sensitive to quantization error in the first token. By keeping only the first token in fp16, we can attain perplexity benefits, particularly for 2-bit quantization. A similar observation has also been made in another concurrent work [23]. Note that when retaining the first token in fp16, we account for this during the calibration process as well, meaning that we ignore the first token when deriving the nuqX datatype and when calibrating the scaling factors and zero points offline for the Keys. As demonstrated in Appendix K, this approach persistently yields performance benefits, particularly with lower bit widths and without dense-and-sparse quantization.

### 3.6 Offline Calibration versus Online Computation

A crucial challenge for activation quantization is that we either need to compute statistics on-the-fly (which is potentially expensive) or else we need to use offline calibration data (which potentially has negative accuracy implications). The challenges with computing scaling factors (and zero-points) online versus offline for both Keys and Values are shown in Figure 5 in Appendix L. In per-channel quantization, it is challenging to update scaling factors online since the scaling factors corresponding to each incoming channel would potentially need to be updated whenever a new token is added to the KV cache. It is therefore desirable to be able to compute statistics offline (i.e., using calibration data before running inference). While this can have negative effects on model accuracy, in Appendix L we show that we can effectively calibrate offline for per-channel quantization, obviating the need for online updates of scaling factors for per-channel quantization. For per-token quantization, it is challenging to calibrate for scaling factors offline due to the presence of outlier Value tokens. It is therefore desirable to be able to compute scaling factors and outlier thresholds online for each incoming token. As shown in Appendix L, we can efficiently compute outlier thresholds online per-token by offloading to the CPU. By leveraging custom quantization function implementations for compressing activations, we are able to perform online per-token Value quantization without compromising on performance.

### 3.7 Kernel Implementation

In order to efficiently perform activation quantization on-the-fly, we leverage dedicated kernel implementations with our 4-bit quantization method for compressing vectors to reduced precision and extracting the sparse outliers, performing matrix-vector multiplications using the compressed

**Table 1:** *Evaluation of our method for different models using the perplexity (PPL) on Wikitext-2. KVQuant results are using pre-RoPE per-channel quantization for Keys. KV cache sizes are estimated assuming a sequence length of 128K (ignoring context length limits for the models). Note that ATOM and FlexGen use 4-bit quantization with group sizes of 128 and 64 with uniform quantization, respectively, and we extend their methods to 3-bit and 2-bit quantization. We leverage Attention Sink-Aware quantization for all bit widths. We used post-RoPE quantization for all baseline methods since it achieves higher accuracy when quantizing Keys per-token as shown in Appendix P. Table 18 in Appendix O demonstrates a full evaluation on all LLaMA, Llama-2, Llama-3, and Mistral models.*

| Method | LLaMA-7B | | LLaMA-13B | | LLaMA-30B | | LLaMA-65B | |
|---|---|---|---|---|---|---|---|---|
| | PPL | KV Cache (GB) | PPL | KV Cache (GB) | PPL | KV Cache (GB) | PPL | KV Cache (GB) |
| baseline | 5.68 | 64.0 | 5.09 | 100.0 | 4.10 | 195.0 | 3.53 | 320.0 |
| int4 | 5.98 | 16.0 | 5.32 | 25.0 | 4.34 | 48.8 | 3.73 | 80.1 |
| nf4 | 5.87 | 16.0 | 5.23 | 25.0 | 4.25 | 48.8 | 3.63 | 80.1 |
| ATOM-4bit | 5.77 | 16.6 | 5.16 | 26.0 | 4.16 | 50.7 | 3.57 | 83.1 |
| FlexGen-4bit | 5.73 | 17.3 | 5.14 | 27.0 | 4.14 | 52.6 | 3.56 | 86.3 |
| KVQuant-4bit | 5.72 | 16.0 | 5.13 | 25.0 | 4.13 | 48.8 | 3.55 | 80.0 |
| KVQuant-4bit-1% | **5.69** | 17.3 | **5.10** | 27.0 | **4.11** | 52.7 | **3.54** | 86.5 |
| int3 | 10.87 | 12.0 | 8.69 | 18.8 | 6.82 | 36.6 | 6.37 | 60.1 |
| nf3 | 7.33 | 12.0 | 6.21 | 18.8 | 5.46 | 36.6 | 4.44 | 60.1 |
| ATOM-3bit | 6.17 | 12.6 | 5.47 | 19.7 | 4.44 | 38.4 | 3.78 | 63.0 |
| FlexGen-3bit | 5.93 | 13.2 | 5.29 | 20.6 | 4.26 | 40.2 | 3.66 | 65.9 |
| KVQuant-3bit | 5.87 | 12.0 | 5.25 | 18.8 | 4.25 | 36.6 | 3.63 | 60.0 |
| KVQuant-3bit-1% | **5.75** | 13.3 | **5.14** | 20.8 | **4.15** | 40.5 | **3.57** | 66.5 |
| int2 | 11779 | 8.0 | 69965 | 12.5 | 1470 | 24.4 | 7272 | 40.1 |
| nf2 | 3210 | 8.0 | 5786 | 12.5 | 2044 | 24.4 | 5367 | 40.1 |
| ATOM-2bit | 37.37 | 8.6 | 41.77 | 13.4 | 16.49 | 26.1 | 13.63 | 42.8 |
| FlexGen-2bit | 11.09 | 9.1 | 9.84 | 14.3 | 6.60 | 27.8 | 5.54 | 45.6 |
| KVQuant-2bit | 7.23 | 8.0 | 5.82 | 12.5 | 4.87 | 24.4 | 4.03 | 40.0 |
| KVQuant-2bit-1% | **6.01** | 9.3 | **5.36** | 14.5 | **4.35** | 28.3 | **3.70** | 46.5 |

vectors, and performing sparse matrix-dense vector multiplications using the sparse outliers. We store the quantized Key and Value matrices as 4-bit elements which are used as indices into lookup tables to recover the corresponding fp16 values. We store the sparse outlier matrices in either Compressed-Sparse Row (CSR) or Compressed-Sparse Column (CSC) format (depending on which aligns better with appending new Key and Value tokens). The kernels for the Key matrix-vector operations apply RoPE on-the-fly in order to support pre-RoPE quantization. More kernel implementation details are provided in Appendix R.

# 4 Results

## 4.1 Main Evaluation

We used the LLaMA-7B/13B/30B/65B, Llama-2-7B/13B/70B, Llama-3-8B/70B, and Mistral-7B models to evaluate our methodology by measuring perplexity on both Wikitext-2 and C4 [36, 37, 1, 16, 27, 31]. Perplexity has been measured using teacher forcing with the output logits of all input tokens. We compared our method against (i) uniform quantization without grouping (intX), (ii) nonuniform quantization using NormalFloat [8] without grouping (nfX), as well as (iii) Atom [44] and FlexGen [34]. Note that Atom and FlexGen use uniform quantization with group sizes of 64 and 128, respectively. All the KVQuant models throughout this experiment section are calibrated using 16 calibration samples of sequence length 2K from the Wikitext-2 training set. See Appendix M for details on our experimental setup, including our methodology for computing KV cache size estimates.

Table 1 shows the results for LLaMA models for the Wikitext-2 dataset. We compared our method with per-token quantization with and without grouping. The baseline configurations used by Atom and FlexGen are included for reference [44, 34]. We find that our method consistently outperforms baseline approaches by an especially large margin with 3-bit and 2-bit quantization. Once we incorporate outliers, we further push the performance of low-precision quantization, achieving 4-bit quantization with less than 0.02 perplexity degradation, 3-bit quantization with under 0.1 perplexity degradation, and 2-bit quantization with under 0.5 perplexity degradation on Wikitext-2, relative to the fp16 baseline, across all models (while attaining 3.7×, 4.8×, and 6.9× memory savings, respectively).

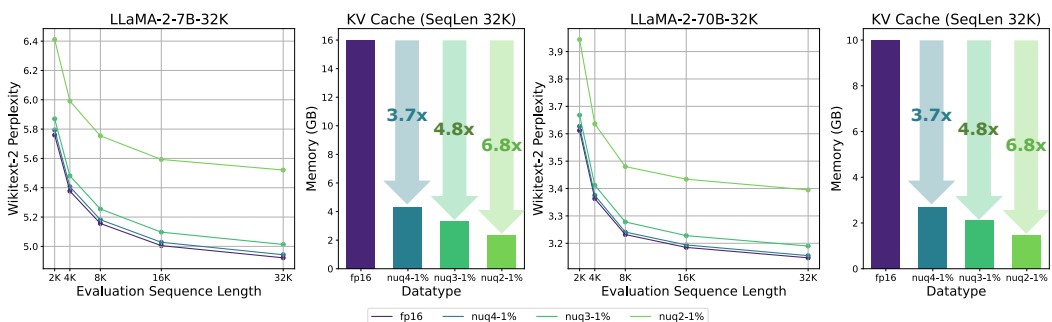

**Figure 3:** *Perplexity results for the LLaMA-2-7B-32K model [5] as well as the Llama-2-70B-32K LongLoRA model [6] on the Wikitext-2 dataset, evaluated using different sequence lengths.*

**Table 2:** *Passkey retrieval results across different context lengths for the LLaMA-2-7B-32K model (uptrained for long sequence lengths using positional interpolation [5]) as well as the Llama-2-70B-32K LongLoRA model [6]. The values reported are the success rate for retrieving the passkey, computed over 50 samples. We also include comparisons with KIVI for reference, using the 2-bit configuration with group size of 32 and 128-element fp16 residual [26]. Average bit widths are estimated for each approach assuming 32K context length. Note that the open-source code for running KIVI with LLaMA does not support grouped-query attention, so we did not include comparisons with KIVI for Llama-2-70B-32K.*

| Model | Method | 2K | 4K | 8K | 16K | 32K | Avg. Bit Width |
|---|---|---|---|---|---|---|---|
| LLaMA-2-7B-32K | fp16 | 1 | 1 | 1 | 1 | 1 | 16 |
| | KIVI-2-gs32-r128 | 0.76 | 0.72 | 0.72 | 0.68 | 0.7 | 3.05 |
| | nuq4-1% | 1 | 1 | 1 | 1 | 1 | 4.33 |
| | nuq3-1% | 0.98 | 1 | 1 | 1 | 1 | 3.33 |
| | nuq2-1% | 1 | 1 | 0.98 | 1 | 1 | 2.33 |
| Llama-2-70B-32K | fp16 | 1 | 1 | 1 | 1 | 1 | 16 |
| | nuq4-1% | 1 | 1 | 1 | 1 | 1 | 4.35 |
| | nuq3-1% | 1 | 1 | 1 | 1 | 1 | 3.35 |
| | nuq2-1% | 0.98 | 0.98 | 0.96 | 1 | 0.74 | 2.35 |

## 4.2 Long Context Length Evaluation

**Perplexity Evaluation.** We evaluated long context length performance using the LLaMA-2-7B-32K model (uptrained for long sequence lengths using positional interpolation [5]) as well as the Llama-2-70B-32K LongLoRA model [6]. For evaluating performance on longer context lengths, we first evaluated perplexity on Wikitext-2 using larger amounts of input context, as shown in Figure 3 [6, 13]. The results demonstrate how our method maintains accuracy even for longer amounts of input context, thereby enabling efficient and accurate long sequence length inference.

**Passkey Retrieval Evaluation.** We also evaluated the performance of our quantization method on passkey retrieval to assess the model's ability to use its context. Passkey retrieval involves evaluating the model's capacity to locate specific information in long texts [19], and this can be used to effectively measure the maximum distance over which a token can attend during the inference stage. We used the passkey evaluation framework from [45] (which is based on the methodology from [28]) to evaluate retrieval performance. The passkey retrieval results are provided in Table 2, demonstrating how our method is able to maintain the retrieval performance for long context length models. We also include comparisons with the passkey retrieval when using KIVI for the LLaMA-2-7B-32K model [26], demonstrating that our approach can attain higher retrieval rate for the same compression level. Our improved performance on retrieval tasks can be attributed to our improved representation of all tokens equally. This approach differs from KIVI, which preserves a local window of residual tokens in fp16. Therefore, while KIVI is effective at representing the tail part of the context, it may provide less benefit for tasks requiring the utilization of the full context window.

**LongBench Evaluation.** Table 3 shows evaluation on LongBench [3] for the LLaMA-2-7B-32K model. LongBench contains a suite of long-context length evaluation benchmarks including QA tasks, summarization, and few-shot learning [3]. The max input context length is set at 31500, and results using KIVI are also included for reference [26]. Our results demonstrate that our 3-bit

**Table 3:** *LongBench evaluation for the LLaMA-2-7B-32K model using KVQuant-3bit-1%. Comparisons with KIVI are included for reference, using the configuration with group size of 32 and 128-element fp16 residual [26]. Average bit widths are estimated for each approach assuming 12.2K context length, which was the average number of tokens across all tasks.*

| Config | Avg. bit | NtrvQA | Qasper | MF-en | Hotpot | 2Wiki | Musique | GovRep | QMSum | MNews | TREC | TriviQA | SamSum | RBench | LCC | PsgRetr | PsgCnt | Avg. |
|---|---|---|---|---|---|---|---|---|---|---|---|---|---|---|---|---|---|---|
| fp16 Baseline | 16 | 17.96 | 10.51 | 33.43 | 12.55 | 12.53 | 6.19 | 29.65 | 16.99 | 22.15 | 71 | 87.79 | 43.97 | 59.99 | 62.14 | 23 | 1.50 | 31.96 |
| KIVI-2-gs32-r128 | 3.17 | **19.25** | 10.66 | 24.78 | **12.48** | 11.19 | **6.38** | 27.05 | 16.36 | 23.37 | 71 | 80.80 | 43.93 | 57.74 | 60.61 | 13.58 | 1.50 | 30.04 |
| KVQuant-3bit-1% | 3.33 | 18.87 | **13.67** | **30.93** | 12.07 | **12.55** | 6.25 | **27.10** | **16.53** | 16.54 | 71 | 87.55 | 43.95 | 59.50 | 61.52 | 19.5 | **1.75** | **31.21** |

**Table 4:** *RULER evaluation results for the LLaMA-2-7B-32K model with KVQuant quantization methods. We report accuracy across RULER tasks, comparing our KVQuant configurations to baseline and KIVI approaches. A maximum context length of 32K is used for evaluation. Our results show that our method retains baseline accuracy even with aggressive quantization and pruning.*

| Config | Avg. bit | Niah1 | Niah2 | Niah3 | MKey1 | MKey2 | MKey3 | MValue | MQuery | VT | CWE | FWE | QA1 | QA2 | Avg. |
|---|---|---|---|---|---|---|---|---|---|---|---|---|---|---|---|
| fp16 Baseline | 16 | 100 | 99.8 | 98.6 | 94 | 68.2 | 11 | 55.95 | 64.5 | 37.88 | 9.64 | 30.4 | 31.6 | 31.6 | 56.40 |
| KIVI-2-gs32-r128 | 3.05 | 76 | 85.6 | 59.6 | 72.6 | 11.4 | 0 | 34.7 | 46.45 | 39.6 | 8.26 | **30.53** | 24.8 | 27.6 | 39.78 |
| KVQuant-3bit-1% | 3.33 | **99.8** | **98.8** | **95.2** | **92.8** | **61.6** | **6.4** | **47.5** | **54.45** | **41.04** | **8.52** | 29.33 | **31.0** | **31.0** | **53.65** |
| KVQuant-2bit-1% | 2.33 | 95.4 | 86.8 | 49.8 | 73.6 | 23.4 | 0 | 16.65 | 22.95 | 22.52 | 5.14 | 24.0 | 26.4 | 28.4 | 36.54 |

model can attain minimal degradation relative to the fp16 baseline, outperforming KIVI for a similar compression level.

**RULER Evaluation.** Finally in Table 4, we provide evaluation of KVQuant and KIVI on the RULER benchmark suite [15] using LLaMA-2-7B-32K. As can be seen in the table, 3-bit KVQuant achieves 14% better score against KIVI with a similar average bit-width. Furthermore, our 2-bit KVQuant achieves similar accuracy to KIVI with 1.5× smaller bit-width.

## 4.3 Joint Weight and KV Cache Quantization

Table 5 provides results for our KV cache quantization method when the weights are also quantized using the methodology in SqueezeLLM [17]. We observe minimal perplexity degradation when leveraging our KV cache quantization approach, even when weights are also quantized to reduced precision. In particular, we observe small 0.02 and 0.1 perplexity degradation of 4-bit and 3-bit weight-only quantization, respectively, when quantizing the KV cache using nuq4-1% for the LLaMA-7B and LLaMA-13B models. These results demonstrate how our method is compatible with existing weight-only quantization methods.

## 4.4 Performance Analysis and Memory Savings

Table 6 shows kernel benchmarking results using a batch size of 1 for the 4-bit dense-and-sparse compression and matrix-vector kernel implementations. We show results across different sequence lengths to assess the performance of the kernels at different points during generation. We report latency benchmarked on an A6000 GPU. The results show that for the Key and Value multiplications, we can achieve 1.2-1.6× and 1.3-1.7× latency savings, respectively, relative to the baseline. We have integrated these kernels into an end-to-end generation pipeline that is able to compress activations dynamically during inference, thereby achieving significant memory savings and allowing for either larger batch sizes or longer sequence lengths.

Appendix A highlights the benefits of KVQuant in supporting longer context lengths through reducing KV cache memory footprint. As shown in Table 8 in Appendix A, our nuq2 method provides 8× KV cache compression and enables serving the quantized LLaMA-7B model with a context length of **1M tokens** on a single A100 GPU, as well as enabling serving the LLaMA-7B model with **10M context length** on an 8-GPU system. Our results show little degradation compared to baseline fp16 inference

**Table 5:** *KV cache quantization results when KVQuant is applied in conjunction with the weight quantization methodology in SqueezeLLM [17]. w4-s45 and w3-s45 for weights refer to the 4-bit and 3-bit dense-and-sparse weight quantization approaches in [17], respectively. See Appendix M for experimental details.*

| Weights | KV Cache | LLaMA-7B | LLaMA-13B | Avg. Bits (KV Cache) |
|---------|----------|----------|-----------|----------------------|
| fp16 | fp16 | 5.68 | 5.09 | 16 |
| w4-s45 | fp16 | 5.77 | 5.17 | 16 |
|  | nuq4-1% | **5.79** | **5.18** | 4.32-4.33 |
| w3-s45 | fp16 | 6.13 | 5.45 | 16 |
|  | nuq3-1% | **6.23** | **5.52** | 3.32-3.33 |

**Table 6:** *Average latency (in microseconds) for the Key and Value nuq4-1% kernels, benchmarked on an A6000 GPU for the LLaMA-2-7B-32K model across different sequence lengths (l). fp16 matrix-vector multiplication latencies are included for reference, and the fp16 Key multiplication time also includes the time to apply RoPE to the newly appended Key vector. Section 3.7 and Appendix R provide additional details for our kernel implementation, Appendix R describes our benchmarking methodology, and Table 22 provides a detailed breakdown of kernel runtime on an A6000 GPU.*

| Activation | Operation | l=2K | l=4K | l=16K |
|------------|-----------|------|------|-------|
| Key | fp16 Matvec | 33.3 | 59.1 | 219.4 |
| Key | nuq4-1% | 25.6 | 39.9 | 126.3 |
| Value | fp16 Matvec | 26.0 | 50.2 | 203.7 |
| Value | nuq4-1% | 22.1 | 37.9 | 124.5 |

while providing significant compression, demonstrating the benefits of our approach for enabling accurate and efficient long sequence length inference.

## 5 Conclusion

As context lengths in LLMs increase, the KV cache activations surface as the dominant contributor to memory consumption. Quantization is a promising approach to reduce the size of KV cache activations, but prior solutions failed to represent activations accurately in ultra-low precisions, such as sub-4-bit. In contrast, we achieve accurate ultra-low precision KV cache quantization. By quantizing Keys per-channel before applying RoPE, we are able to better match the outlier distribution and mitigate the impacts of RoPE on quantization (due to it mixing pairs of channels which may have different average magnitudes). We use non-uniform quantization to better allocate the small number of quantization signposts at low precision. We observe significant accuracy improvements when employing dense-and-sparse quantization, particularly when detecting outliers at the same granularity as we compute quantization scaling factors. Crucially, we demonstrate that we can perform accurate calibration offline for Keys, as well as efficient online scaling factor and outlier threshold computation for Values. By leveraging these methods, we are able to enable accurate low-precision activation quantization, achieving 4.8x compression (nuq3-1% outliers) with only 0.1 perplexity degradation across different LLaMA, Llama-2, Llama-3, and Mistral models. Our methodology therefore supports inferring the LLaMA-7B model with a context length of **10M** on an 8-GPU serving system. Through our efficient kernel implementation, we are able to show improved latency relative to the fp16 baseline, demonstrating how our method allows for improved latency in addition to the memory savings.

## Acknowledgements

The authors would like to acknowledge Nicholas Lee for helpful discussions and feedback. We acknowledge gracious support from Intel, Furiosa, Apple, Samsung SAIT, and NVIDIA. We also appreciate the support from Microsoft through their Accelerating Foundation Model Research, including great support from Sean Kuno. Furthermore, we appreciate support from Google Cloud, the Google TRC team, and specifically Jonathan Caton, and Prof. David Patterson. Prof. Keutzer's lab is sponsored by the Intel corporation, Intel One-API, Intel VLAB team, the Intel One-API center

of excellence, as well as funding through BDD and BAIR. We appreciate great feedback and support from Ellick Chan, Saurabh Tangri, Andres Rodriguez, and Kittur Ganesh. Sehoon Kim would like to acknowledge the support from the Korea Foundation for Advanced Studies (KFAS). Michael W. Mahoney would also like to acknowledge a J. P. Morgan Chase Faculty Research Award as well as the DOE, NSF, and ONR. Our conclusions do not necessarily reflect the position or the policy of our sponsors, and no official endorsement should be inferred.

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

# A    Memory Bottlenecks for Long Context Length Inference

Table 7 shows the model size and KV cache memory requirements for different LLaMA models with different sequence lengths. For short sequence lengths, the model weights are the primary memory bottleneck. However, for longer sequence lengths and larger batch sizes, the KV cache memory is the main bottleneck. This is particularly pronounced when the weights are already quantized to low precision. In our work, we demonstrate that we can help address the KV cache memory bottleneck through low-precision KV cache quantization. By compressing the KV cache to 2-bit precision, we can enable **1M context length inference with the LLaMA-7B model on a single A100-80GB GPU**, and we can also enable **10M context length inference with the LLaMA-7B model on an 8-GPU system**.

Table 8 shows the KV cache memory requirements for 128K, 1M, and 10M sequence lengths, with the KV cache stored in fp16 as well as 4-bit, 3-bit, and 2-bit precision with KVQuant. As one can see, our method provides $3.7\times$ KV cache compression (nuq4-1%) and enables serving the quantized LLaMA-65B model with a context length of 32K tokens on a single A100-80GB GPU (requiring 30.3GB for the model weights compressed to 4-bit, and 46.5GB for the KV cache when compressed with nuq2-1%), and our nuq2 method enables serving the LLaMA-7B model with a context length of **1M tokens** on a single A100 GPU (requiring 64GB for the KV cache). Additionally, when considering an 8-GPU serving system, we enable serving the LLaMA-7B model with **10M context length** (with nuq2), or the LLaMA-65B model with 1M context length (with nuq3). Our results show little degradation compared to baseline fp16 inference while providing significant compression, demonstrating the benefits of our approach for enabling accurate and efficient long sequence length inference.

**Table 7:** *Model size and activation memory size estimates for different sequence lengths and batch sizes (BS) for different LLaMA models. For long sequence lengths and larger batch sizes, activation memory is the main bottleneck (particularly when weights are already quantized to low precision). By compressing the KV cache to 2-bit precision, we can enable **1M context length inference with the LLaMA-7B model on a single A100-80GB GPU**, and we can also enable **10M context length inference with the LLaMA-7B model on an 8-GPU system**.*

| BS | Model | Model Size (GB) $16 \rightarrow$ 2-bit | KV Cache Size w/ Diff. Seq Len (GB) | | | |
|---|---|---|---|---|---|---|
| | | | 32K | 128K | 1M | 10M (16 $\rightarrow$ 2-bit) |
| 1 | 7B | 12.6 $\rightarrow$ 1.6 | 16 | 64 | 512 | 4883 $\rightarrow$ 610 |
| | 13B | 24.1 $\rightarrow$ 3.0 | 25 | 100 | 800 | 7629 $\rightarrow$ 954 |
| | 30B | 60.3 $\rightarrow$ 7.5 | 49 | 195 | 1560 | 14877 $\rightarrow$ 1860 |
| | 65B | 121.1 $\rightarrow$ 15.1 | 80 | 320 | 2560 | 24414 $\rightarrow$ 3052 |
| 4 | 7B | 12.6 $\rightarrow$ 1.6 | 64 | 256 | 2048 | 19531 $\rightarrow$ 2441 |
| | 13B | 24.1 $\rightarrow$ 3.0 | 100 | 400 | 3200 | 30518 $\rightarrow$ 3815 |
| | 30B | 60.3 $\rightarrow$ 7.5 | 195 | 780 | 6240 | 59509 $\rightarrow$ 7439 |
| | 65B | 121.1 $\rightarrow$ 15.1 | 320 | 1280 | 10240 | 97656 $\rightarrow$ 12207 |

# B    Additional Related Works

## B.1    Outlier-Aware LLM Quantization

LLMs have been known to have distinct outliers both in weights and activations [7, 9, 17]. SqueezeLLM and SpQR both decompose the weight matrix into a sparse matrix containing a small portion of outliers and a dense matrix that can be accurately quantized to low precision (referred to as dense-and-sparse or sparse-quantized representation) [9, 17]. LLM.int8() [7] handled particular outlier channels separately in higher precision, and SmoothQuant [41] migrates quantization difficulty due to outlier channels to weights in order to support joint weight-activation quantization. Other works reconsidered the dimension along which we quantize in order to reduce quantization error (or else added per-channel compensation to improve quantization performance) [4, 14, 39, 38]. In this work, we demonstrate that per-channel pre-RoPE Key quantization provides significant accuracy benefits given the outlier structure in Keys, and that dense-and-sparse quantization can be efficiently applied for KV cache quantization.

**Table 8:** *Activation memory size estimates (GB) for 128K, 1M, and 10M sequence length (l) for different LLaMA models. By compressing the KV cache to 2-bit precision, we can enable 1M context length inference with the LLaMA-7B model on a single A100-80GB GPU, and we can also enable 10M context length inference with the LLaMA-7B model on an 8-GPU system.*

| Model | Method | l=128K | l=1M | l=10M |
|---|---|---|---|---|
| | fp16 | 64.0 | 512.0 | 4882.8 |
| | nuq4 | 16.0 | 128.1 | 1221.9 |
| | nuq4-1% | 17.3 | 138.4 | 1319.6 |
| LLaMA-7B | nuq3 | 12.0 | 96.1 | 916.7 |
| | nuq3-1% | 13.3 | 106.4 | 1014.4 |
| | nuq2 | 8.0 | 64.1 | 611.5 |
| | nuq2-1% | 9.3 | 74.4 | 709.2 |
| | fp16 | 320.0 | 2560.0 | 24414 |
| | nuq4 | 80.0 | 640.3 | 6106.5 |
| | nuq4-1% | 86.5 | 691.5 | 6595.0 |
| LLaMA-65B | nuq3 | 60.0 | 480.3 | 4580.6 |
| | nuq3-1% | 66.5 | 531.5 | 5069.1 |
| | nuq2 | 40.0 | 320.3 | 3054.7 |
| | nuq2-1% | 46.5 | 371.5 | 3543.3 |

## B.2 Non-uniform LLM Quantization

Non-uniform quantization has also been applied in the context of LLMs. Non-uniform quantization allows for more flexible quantization signpost placement relative to uniform quantization methods, enabling improved accuracy for the same bit precision [17, 8]. Building on the observation that model parameters tend to be approximately normally-distributed, prior work has proposed the NormalFloat datatype [8]. SqueezeLLM [17] derived per-channel non-uniform quantization signposts using a sensitivity-weighted k-means approach. In this work, we show that we can derive accurate per-layer non-uniform datatypes using a sensitivity-weighted k-means approach with KV cache activations.

## C  RoPE Equation

The rotation matrix for RoPE is provided in Equation 2, where c and s are cosine and sine functions, $\theta_i = 10000^{-2(i-1)/d}$, $d$ is the attention head dimension, and $n$ is the current position in the sequence:

$$
\begin{bmatrix}
c(n\theta_1) & -s(n\theta_1) & \cdots & 0 & 0 \\
s(n\theta_1) & c(n\theta_1) & \cdots & 0 & 0 \\
\vdots & \vdots & \ddots & \vdots & \vdots \\
0 & 0 & \cdots & c(n\theta_{d/2}) & -s(n\theta_{d/2}) \\
0 & 0 & \cdots & s(n\theta_{d/2}) & c(n\theta_{d/2})
\end{bmatrix}
\tag{2}
$$

The Query vectors computed at each iteration will have RoPE applied (to obtain $\tilde{Q}_m = R^d_{\theta,m} * Q_m$). When caching Key vectors, we therefore need to either cache $\tilde{K}_n = R^d_{\theta,n} * K_n$, or else we need to cache $K_n$ and apply $R^d_{\theta,n}$ on-the-fly during inference. In order to apply $R^d_{\theta,n}$ efficiently on-the-fly, we leverage the element-wise formulation of RoPE rather than the matrix-multiplication formulation from Equation 2. The element-wise formulation for $R^d_{\theta,n}x$ is as follows, where $\odot$ is the element-wise multiplication operator (note that the formulation that we use matches the implementation in the Transformers library for LLaMA [40], and it is a different but equivalent formulation to the element-wise expression in [35]):

$$
\begin{bmatrix} x_1 \\ x_2 \\ \vdots \\ x_{\frac{d}{2}} \\ x_{\frac{d}{2}+1} \\ \vdots \\ x_{d-1} \\ x_d \end{bmatrix} \odot \begin{bmatrix} \mathrm{c}(\theta_1 n) \\ \mathrm{c}(\theta_2 n) \\ \vdots \\ \mathrm{c}(\theta_{\frac{d}{2}} n) \\ \mathrm{c}(\theta_1 n) \\ \vdots \\ \mathrm{c}(\theta_{\frac{d}{2}-1} n) \\ \mathrm{c}(\theta_{\frac{d}{2}} n) \end{bmatrix} + \begin{bmatrix} -x_{\frac{d}{2}+1} \\ -x_{\frac{d}{2}+2} \\ \vdots \\ -x_d \\ x_1 \\ \vdots \\ x_{\frac{d}{2}-1} \\ x_{\frac{d}{2}} \end{bmatrix} \odot \begin{bmatrix} \mathrm{s}(\theta_1 n) \\ \mathrm{s}(\theta_2 n) \\ \vdots \\ \mathrm{s}(\theta_{\frac{d}{2}} n) \\ \mathrm{s}(\theta_1 n) \\ \vdots \\ \mathrm{s}(\theta_{\frac{d}{2}-1} n) \\ \mathrm{s}(\theta_{\frac{d}{2}} n) \end{bmatrix} \tag{3}
$$

By leveraging this element-wise implementation, we can apply RoPE on-the-fly to the Key activations (after dequantizing the Key activations and before multiplying them with the corresponding elements in the Query vector).

## D   Derivation for Sensitivity Analysis

To compute the sensitivity analysis we largely follow the derivation in [29], which was originally provided to compute sensitivity of a NN based classifier but can be extended with minor modifications for quantization.

In particular, the sensitivity measure is based on how much the loss output of the model is perturbed. To compute this we denote activations at a layer before quantization as $\mathbf{A}$, after quantization as $\mathbf{A_Q}$, quantization perturbation in activation as $\mathbf{A} - \mathbf{A_Q}$, and the gradient of the Loss function w.r.t. activation as $J(A) = \frac{\partial \mathcal{L}}{\partial A}(A)$. By making the assumption that the quantization perturbation of different activations follow a Gaussian distribution with zero mean, we can show that the sensitivity of an activation value is proportional to $\mathcal{F}_{ii}\big(A - Q(A)\big)^2$ as was given in Equation 1:

$$
\mathbb{E}_{\mathbf{\Delta A}}\left[ |\mathcal{L}(\mathbf{A}) - \mathcal{L}(\mathbf{A} + \mathbf{\Delta A})|^2 \right] \approx \mathbb{E}_{\mathbf{\Delta A}}\left[ \left(\mathbf{J}(\mathbf{A})^T \mathbf{\Delta A}\right)^2 \right] = \mathbb{E}_{\mathbf{\Delta A}}\left[ \left( \sum_i J_i \Delta A_i \right)^2 \right]
$$

$$
= \mathbb{E}_{\mathbf{\Delta A}}\left[ \sum_i J_i^2 \Delta A_i^2 \right]
$$

$$
= \sum_i J_i^2 \mathbb{E}_{\mathbf{\Delta A}}\left[ \Delta A_i^2 \right].
$$

Here note that we first assume a first order Taylor series expansion to approximate the perturbation to the loss, and then use the zero mean assumption of the quantization perturbation to derive the second line. To approximate the $\mathbb{E}_{\mathbf{\Delta A}}\left[ \Delta A_i^2 \right]$ we use empirical evaluation of the expectation by sampling multiple different inputs and empirically computing the resulting quantization perturbation which results in Equation 1.

## E   Derivation for Quantization Error

In our work, before applying the sensitivity-weighted K-means to derive quantization signposts, we normalize each element $A_i$ in the flattened activation $A$. This normalization for $A_i$ involves a shift by a zero-point $z_i$ followed by rescaling the quantization signposts by a scaling factor $s_i$, where $s_i$ and $z_i$ are the scaling factor and zeropoint corresponding to element $A_i$:

$$
A_{i,norm} = \frac{A_i - z_i}{s_i}, \tag{4}
$$

where $A_i$ and $A_{i,norm}$ are element $i$ from activation $A$ before and after normalization, respectively. We then quantize $A_{i,norm}$ to $Q(A_{i,norm})$ with quantization error $\Delta A_{i,norm}$. After we dequantized, we rescale each element by $s_i$ and add $z_i$ to get the recovered quantized activation value $Q(A_i)$:

$$Q(A_i) = s_i \, Q(A_{i,norm}) + z_i. \tag{5}$$

As such, if there is quantization error $\Delta A_{i,norm}$ in $A_{i,norm}$, this will be scaled by $s_i$ in terms of the error in $A_i$, i.e., $\Delta A_i = s_i \Delta A_{i,norm}$. For activation $A$ which is normalized to $A_{norm}$ (with corresponding scaling factors $s_i$ for each element $A_i$), minimizing the sensitivity-weighted quantization error as expressed in Equation 1 gives us the following expression, which we can minimize using the normalized activations across all $N$ elements from the samples in a calibration set:

$$Q(A)^* \simeq \arg\min_Q \sum_{i=1}^{N} \mathcal{F}_{ii}\big(A_i - Q(A_i)\big)^2 \tag{6}$$

$$= \arg\min_Q \sum_{i=1}^{N} \mathcal{F}_{ii}\Big(s_i^2\big(A_{i,norm} - Q(A_{i,norm})\big)^2\Big) \tag{7}$$

## F Key and Value Dynamic Range

Figure 4 shows the portion of the elements contained within difference percentages of the dynamic range for both Keys and Values. The majority of values ($\sim 99\%$) are contained in a small portion of the dynamic range, and a small portion of numerical outliers skew the dynamic range that must be represented. This motivates our dense-and-sparse approach which removes numerical outliers and stores them in a separate sparse matrix, thereby restricting the range that needs to be represented in the dense component.

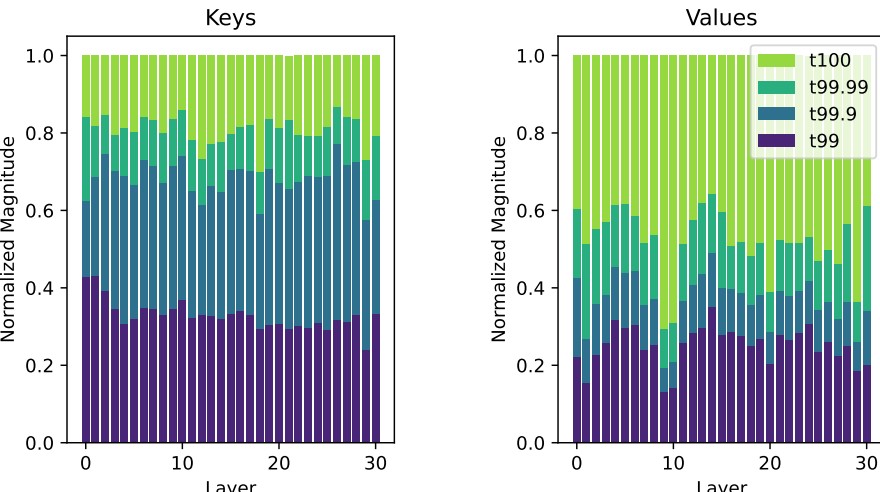

**Figure 4:** *Distribution of the magnitude of elements of Key (Pre-RoPE) and Value activations for different layers of LLaMA-7B, computed on a single sample with sequence length 2K from the Wikitext-2 dataset. The normalized magnitude is computed by dividing by the largest magnitude value in that layer. As one can see, for both Key and Value activations, the majority of values lie in a small portion of the dynamic range, with a few numerical outliers skewing the dynamic range (and thereby reducing the fidelity when quantizing to low precision).*

## G Per-Channel Key Quantization Ablations

As shown in Table 9, per-channel quantization for Keys and per-token quantization for Values outperforms the standard per-token quantization approach for both Keys and Values, yielding an improvement of 3.82 perplexity for the LLaMA-7B model at 3-bit precision. This demonstrates the

benefits of per-channel Key quantization to mitigate the large outlier channels in Keys. Note that for all experiments using per-channel quantization, we use an fp16 zeropoint rather than a low-precision zeropoint that is rounded to the nearest integer value. We do this since for some of the Key channels, all of the elements are positive or all of the elements are negative, meaning that the zeropoint will fall outside of the range that is representable by a low-precision integer (and rounding it to the nearest low-precision value can degrade performance).

Additionally, we observe that per-channel quantization for Values actually performs worse than per-token quantization. We hypothesize that this behavior is because per-channel Value quantization leads to greater error accumulation in particular output values (since the result of the attention scores multiplied by one channel of the Values will be localized to a single value in the output vector), which leads to greater quantization error at later model layers. Another concurrent work, KIVI [26], observes similar behavior for per-channel Value quantization, which they attribute to the fact that per-token Value quantization confines the error to each token. Assuming that the output is a weighted sum of only a few important tokens (as only a few attention scores are large), a perturbation in these tokens can lead to significant degradation. Per-token Value quantization therefore ensures that the quantization of unimportant tokens does not adversely impact the important tokens.

**Table 9:** *Ablation Study: Perplexity comparison of per-token and per-channel quantization for KV cache activations for LLaMA-7B. PT refers to per-token quantization, and PC refers to per-channel quantization.*

| Datatype | Key Dim. | Value Dim. | Perplexity | KV Cache Size (GB) Seqlen 128K |
|----------|----------|------------|------------|--------------------------------|
| fp16 | - | - | 5.68 | 64.0 |
| int3 | PT | PT | 10.87 | 12.0 |
| int3 | PC | PC | 223 | 12.0 |
| int3 | PC | PT | **7.05** | 12.0 |

# H  Pre-RoPE Key Quantization Ablations

As shown in Table 10, pre-RoPE Key quantization achieves higher accuracy than post-RoPE quantization, with an improvement of 0.82 perplexity for 3-bit quantization with the LLaMA-7B model. These results show that the rotary positional embeddings make Key quantization more challenging due to mixing pairs of channels with different magnitudes. Pre-RoPE quantization thereby allows for more accurate quantization at low precision.

**Table 10:** *Ablation Study: Perplexity comparison of Pre-RoPE and post-RoPE Key quantization for LLaMA-7B (using per-channel Key quantization and per-token Value quantization). Pre-RoPE quantization leads to significant improvement (see Section 3.2 for more details).*

| Datatype | Scheme | Perplexity | KV Cache Size (GB) Seqlen 128K |
|----------|--------|------------|--------------------------------|
| fp16 | - | 5.68 | 64.0 |
| int3 | post-RoPE | 7.05 | 12.0 |
| int3 | pre-RoPE | **6.23** | 12.0 |

# I  Sensitivity-Weighted Non-Uniform Quantization Ablations

Table 11 shows perplexity evaluation results across different LLaMA, Llama-2, and Mistral models on Wikitext-2 for different datatypes, including nuq3, nuq3 without using sensitivity-weighting, as well as nuq3 with sensitivity-weighting but without accounting for the per-channel scaling factors when performing k-means. We observe particularly noticeable gains relative to uniform quantization for 3-bit and 2-bit quantization, where the benefits of non-uniform quantization are more pronounced due to the reduced precision. These results demonstrate the benefits of our sensitivity-weighted non-uniform quantization approach relative to NormalFloat quantization [8], as we achieve consistent accuracy improvements of up to 0.33 perplexity across different models. These results also demonstrate the

necessity of our sensitivity-weighting approach in order to derive performant non-uniform datatypes using a k-means based approach. Additionally, we observe distinct benefits when also accounting for the per-channel scaling factors when performing k-means.

**Table 11:** *Ablation Study: Ablation of our sensitivity-weighted non-uniform datatype for different models on Wikitext-2. All experiments use pre-RoPE per-channel quantization for Keys and per-token quantization for Values (meaning that all configurations are the same as in KVQuant, except for the datatype). We compare against both uniform (int3) and non-uniform (nf3) [8] approaches, as well as with using "unweighted" k-means (i.e., not sensitivity-weighted) and "Fisher-weighted k-means" (without accounting for per-channel scaling factors) to compute the non-uniform quantization signposts. Note that there is slight variation in average bitwidth across models due to the differing hidden dimensions. Results report perplexity with 2K/4K/8K sequence length for LLaMA, Llama-2, and Mistral, respectively.*

| Method | LLaMA | | | | Llama-2 | | | Mistral-7B | Avg. Num. Bits |
| | 7B | 13B | 30B | 65B | 7B | 13B | 70B | | |
|---|---|---|---|---|---|---|---|---|---|
| baseline | 5.68 | 5.09 | 4.10 | 3.53 | 5.12 | 4.57 | 3.12 | 4.76 | 16 |
| int3 | 6.23 | 5.60 | 5.09 | 5.18 | 5.95 | 5.98 | 3.26 | 5.26 | 3.00-3.02 |
| nf3 | 6.05 | 5.42 | 4.51 | 3.84 | 5.55 | 5.15 | 3.27 | 5.13 | 3.00-3.02 |
| nuq3 (unweighted) | 6.84 | 6.16 | 5.37 | 4.57 | 8.52 | 7.66 | 3.67 | 5.29 | 3.00-3.02 |
| nuq3 (Fisher-weighted) | 6.01 | 5.34 | 4.41 | 3.74 | 5.49 | 4.83 | 3.26 | 5.03 | 3.00-3.02 |
| nuq3 (KVQuant) | **5.94** | **5.32** | **4.34** | **3.68** | **5.39** | **4.82** | **3.23** | **4.98** | 3.00-3.02 |

## J Per-Vector Dense-and-Sparse Quantization Ablations

Table 12 shows the performance improvements we observe when isolating a small portion of outliers and storing them in a sparse format. We provide results both with using a single per-matrix outlier threshold, as well as with applying separate outlier thresholds per-vector. In particular, we see greater improvements by employing outlier detection with a different threshold per-channel for Keys and per-token for Values. This provides additional benefits since some values which would be considered outliers for the entire matrix are not actually outliers within a particular channel (so they are not hard to quantize). It is therefore better to directly target the outliers that will skew the quantization range for a particular channel. By removing 1% of outliers using per-vector thresholds, we can achieve an additional 0.19 reduction in perplexity for the LLaMA-7B model at 3 bits, **thereby enabling 3-bit quantization with under 0.1 degradation in perplexity**.

**Table 12:** *Ablation Study: Perplexity comparison of different outlier isolation methods for LLaMA-7B on Wikitext-2. Per-vector outlier detection allows for significant accuracy improvements relative to per-tensor outlier detection. All experiments use per-token quantization for Values and per-channel quantization for Keys (pre-RoPE). "PV" refers to using per-vector outlier thresholds, and "PM" refers to using a single per-matrix outlier threshold.*

| Datatype | % Outliers | Outlier Dim. | Perplexity | KV Cache Size (GB) Seqlen 128K |
|---|---|---|---|---|
| fp16 | - | - | 5.68 | 64.0 |
| nuq3 | - | - | 5.94 | 12.0 |
| nuq3 | 0.1% | PM | 5.89 | 12.2 |
| nuq3 | 0.1% | PV | **5.82** | 12.2 |
| nuq3 | 1% | PM | 5.85 | 13.3 |
| nuq3 | 1% | PV | **5.75** | 13.3 |

## K Attention Sink-Aware Quantization Ablations

Table 13 provides perplexity with and without Attention Sink-Aware quantization across different LLaMA, Llama-2, and Llama-3 models. For all bit widths (4, 3, and 2-bit) and in both dense-only and dense-and-sparse quantization settings, Attention Sink-Aware quantization consistently yields perplexity improvement. Notably, the perplexity gain is more pronounced at lower bit widths

and without sparsity. We observe particularly significant improvements for Llama-3 models, with perplexity improvements of 0.25 and 0.13 PPL for nuq2-1% on Llama-3-8B and Llama-3-70B, respectively.

**Table 13:** *Ablation Study: Perplexity with and without Attention Sink-Aware quantization* on *Wikitext-2 with various models. Attention Sink-Aware quantization consistently improves perplexity across different models and bit widths, particularly with lower bit widths and without sparsity.*

| Datatype | Attention Sink-Aware | LLaMA-7B | LLaMA-13B | Llama-2-7B | Llama-2-13B | Llama-3-8B | Llama-3-70B |
|---|---|---|---|---|---|---|---|
| fp16 | - | 5.68 | 5.09 | 5.12 | 4.57 | 5.54 | 2.59 |
| nuq4 | X | 5.72 | 5.14 | 5.17 | 4.62 | 5.64 | 2.65 |
| nuq4 | O | 5.72 | 5.13 | 5.16 | 4.60 | 5.60 | 2.62 |
| nuq4-1% | X | 5.69 | 5.10 | 5.13 | 4.59 | 5.57 | 2.60 |
| nuq4-1% | O | 5.69 | 5.10 | 5.13 | 4.58 | 5.56 | 2.60 |
| nuq3 | X | 5.94 | 5.32 | 5.39 | 4.82 | 6.10 | 2.99 |
| nuq3 | O | 5.87 | 5.25 | 5.34 | 4.71 | 5.84 | 2.72 |
| nuq3-1% | X | 5.75 | 5.15 | 5.18 | 4.62 | 5.67 | 2.66 |
| nuq3-1% | O | 5.75 | 5.14 | 5.17 | 4.61 | 5.64 | 2.63 |
| nuq2 | X | 8.47 | 7.29 | 11.20 | 23.34 | 16.63 | 6.79 |
| nuq2 | O | 7.23 | 5.82 | 7.03 | 9.59 | 7.04 | 3.49 |
| nuq2-1% | X | 6.05 | 5.39 | 5.47 | 4.85 | 6.29 | 2.95 |
| nuq2-1% | O | 6.01 | 5.36 | 5.41 | 4.78 | 6.04 | 2.82 |

## L  Calibration Ablations

Figure 5 outlines the accuracy and efficiency challenges which our work addresses in order to enable accurate and efficient KV cache quantization. We demonstrate that we can circumvent the challenges related to efficiently performing online per-channel scaling factor computation by instead calibrating offline for the scaling factor without hurting accuracy. Additionally, we show that we can perform per-token scaling factor and outlier threshold computation efficiently online during inference, thereby enabling accurate per-token quantization without compromising on efficiency.

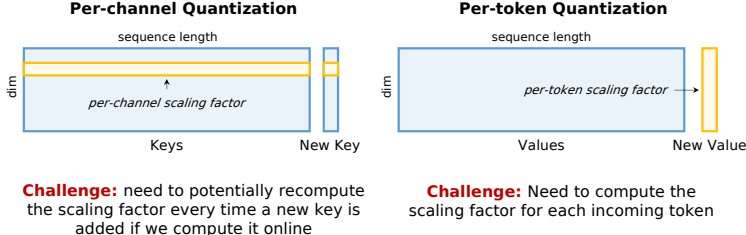

**Figure 5:** *One typically achieves better performance when the scaling factor/zero point are computed online. However, this is quite challenging to do for per-channel quantization, as these factors will not only need to be recomputed for every new Key appended to the Key cache, but also all the prior cached Keys will need to be updated. As such, we use a calibration set to compute per-channel scaling factors offline. A similar challenge exists for per-token quantization, but online calibration for this does not require updating prior cached Values. In Section 3.6 and Appendix L, we discuss how we are able to efficiently compute outlier thresholds / scaling factors for per-token calibration, thereby enabling online computation.*

Table 14 shows accuracy results when using offline calibration for computing the scaling factors for the Keys. For 3-bit quantization, we observe minor accuracy degradation when not employing outlier extraction methods. However, if we remove a small percentage of outliers, then the accuracy with offline calibration is the same as computing the scaling factors online per-channel during evaluation. This demonstrates that when incorporating outlier extraction methods, we are better able to perform offline calibration due to reduced sensitivity to outliers (either to outliers during calibration that exaggerate the quantization range, or to outliers during evaluation that cannot be represented accurately if there weren't large outliers observed during calibration).

Table 15 shows the runtime for the $topk$ operation for the LLaMA-7B model (which is required for computing outlier thresholds online). It compares the runtime of the $topk$ operation with the runtime for the QKV projections, finding that the $topk$ runtime is 45% of the matrix-vector operation runtime for a single projection layer. The $topk$ operation can also be performed efficiently on the CPU, so we can actually run this operation *in parallel* with the subsequent linear layer matrix-vector operations on the GPU (which is possible by computing the Value projection before the Key and Query projections). This allows us to compress the activations dynamically without added runtime overhead, thereby enabling online scaling factor computation for the Value tensors.

Table 16 shows how both Fisher information computation and calibration (including k-means) per-layer take only a few minutes for the LLaMA-65B model on a typical server machine. Even if we perform calibration sequentially for each layer, the entire calibration process would take a maximum of 6 hours for the LLaMA-65B model at 4-bit precision.

**Table 14:** *Ablation Study: Model accuracy when using offline calibration for Keys* with LLaMA-7B. *When incorporating outlier detection, offline calibration for Keys is able to perform comparably with online calibration. All nuq3 experiments use per-token quantization for Values and per-channel quantization for Keys (pre-RoPE), and experiments with outliers use per-vector outlier detection.*

| Datatype | % Outliers | Perplexity (Online for K) | Perplexity (Offline for K) |
|---|---|---|---|
| fp16 | - | 5.68 | 5.68 |
| nuq3 | - | 5.91 | 5.94 |
| nuq3 | 1% | 5.75 | 5.75 |

**Table 15:** $topk$ *runtime on a vector of length 4096 for computing outlier thresholds when using 1% sparsity (compared with the runtime for the QKV matrix multiplications, which are 4096×4096 by 4096 matrix-vector multiplications for the LLaMA-7B model). The runtime is reported on a system with an A6000 GPU and an Intel Xeon Gold 6126 CPU. We find that the runtime for the $topk$ operation is only 45% of the runtime of each matvec operation. Additionally, the $topk$ operation can be performed efficiently on the CPU; we can therefore run this operation in parallel with subsequent linear layer operations on the GPU to compress the activations dynamically without added overhead.*

| Operation | Device | Outlier % | Runtime (ms) |
|---|---|---|---|
| QKV Projection | GPU | - | 0.172 |
| $topk$ | CPU | 1% | 0.026 |
| $topk$ | GPU | 1% | 0.088 |
| QKV Projection / $topk$ (Fused) | GPU / CPU | 1% | 0.173 |

**Table 16:** *Runtime for computing Fisher information as well as for calibration (including k-means) with 16 samples for LLaMA-65B quantization. Runtime for computing Fisher information was computed on an 8-GPU A100-80GB system. Runtime for calibration (including k-means) was performed on an Intel Xeon Gold 6442Y CPU, and is shown for a single layer. Note that calibration is independent for each layer, so it can be easily parallelized.*

| Operation | Runtime (minutes) |
|---|---|
| Computing Fisher Information | 2.8 |
| 4-bit Calibration Per-Layer (including k-means) | 4.5 |
| 3-bit Calibration Per-Layer (including k-means) | 2.7 |
| 2-bit Calibration Per-Layer (including k-means) | 1.9 |

# M  Additional Experimental Details

For our empirical evaluation, we use 16 calibration samples of sequence length 2K from the Wikitext-2 training set (as well as the corresponding gradients) to derive the per-channel scaling factors and zero-points, and to derive the non-uniform datatypes for both Keys and Values. While we use KVQuant models that are calibrated on the Wikitext-2 dataset for all experiments, in Appendix L, we

include an additional experiment that demonstrates the robustness of the calibration process to the choice of data.

We measured perplexity on both Wikitext-2 and on C4 using a sequence length equal to the maximum context length of the model (2K for LLaMA, 4K for Llama-2, and 8K for Llama-3 and Mistral-7B). For generative tasks, when processing the input prompt, the Key/Value matrix multiplications are computed using the fp16 Keys and Values, and then they are separately compressed into low precision. For baseline experiments, we use post-RoPE quantization, both since this is required from an efficiency perspective without a dedicated kernel implementation, and because it provides better accuracy when quantizing Keys per-token as shown in Appendix P.

We make several assumptions in order to estimate average bit widths and KV cache sizes for different approaches. We compute these estimates assuming a sequence length of 128K (unless otherwise specified). For integer quantization, we assume a low-precision integer offset and a 16-bit scaling factor, whereas for NormalFloat and NUQ we assume that the zero-point and offset are each 16-bit. For the sparse matrices, 32-bit integers are assumed for the per-token indices (since we need to support long sequence lengths), and the elements and per-element indices are assumed to be 16-bit. This means that for CSR, the rows are assumed to be 32-bit and the columns and values are assumed to be 16-bit, whereas for CSC, the columns are assumed to be 32-bit and the rows and values are assumed to be 16-bit.

## N    Comparison Between Different Datatypes and Sparsity Levels

Table 17 shows perplexity across LLaMA, Llama-2, Llama-3, and Mistral models on Wikitext-2 using different datatypes. The results highlight that the NUQ datatype generally outperforms both uniformly quantized INT datatype and non-uniformly quantized NF datatype even after they are incorporated with grouping at the expense of increased bit-widths. This performance can be further improved by extracting 0.1% to 1.0% of outliers. Note that NUQ with a sparsity level 1.0% has a similar memory requirement as INT with a group size of 64, but achieves significant perplexity improvement across all models and bit widths.

## O    Full Perplexity Evaluation

Tables 18 and 19 show perplexity evaluation across all LLaMA, Llama-2, Llama-3, and Mistral models on Wikitext-2 and C4, respectively. These results demonstrate the benefits of our approach for KV cache compression across different models and bit widths.

**Table 17:** *Comparison of our NUQ datatype with and without sparsity against other data types on different models using the perplexity (PPL) measured on Wikitext-2. Non-uniform quantization ("nuq") results are using pre-RoPE per-channel quantization for Keys. "gs64/128" refers to baseline experiments using grouping with group size 64/128. Note that there is a slight variation in average bitwidth across models due to the differing hidden dimensions.*

| Method | LLaMA 7B | 13B | 30B | 65B | Llama-2 7B | 13B | 70B | Llama-3 8B | 70B | Mistral-7B | Avg. Num. Bits |
|---|---|---|---|---|---|---|---|---|---|---|---|
| baseline | 5.68 | 5.09 | 4.10 | 3.53 | 5.12 | 4.57 | 3.12 | 5.54 | 2.59 | 4.76 | 16 |
| int4 | 5.98 | 5.32 | 4.34 | 3.73 | 5.66 | 5.01 | 3.31 | 7.89 | 14.03 | 4.97 | 4.00-4.02 |
| int4-gs128 | 5.77 | 5.16 | 4.16 | 3.57 | 5.32 | 4.71 | 3.16 | 5.78 | 2.73 | 4.82 | 4.16 |
| int4-gs64 | 5.73 | 5.14 | 4.14 | 3.56 | 5.25 | 4.66 | 3.14 | 5.68 | 2.66 | 4.80 | 4.31 |
| nf4 | 5.87 | 5.23 | 4.25 | 3.63 | 5.47 | 4.90 | 3.22 | 6.42 | 3.86 | 4.91 | 4.00-4.03 |
| nf4-gs128 | 5.77 | 5.17 | 4.17 | 3.58 | 5.30 | 4.71 | 3.16 | 5.84 | 2.75 | 4.83 | 4.25 |
| nuq4 | 5.72 | 5.14 | 4.14 | 3.56 | 5.17 | 4.62 | 3.14 | 5.64 | 2.65 | 4.81 | 4.00-4.02 |
| + 0.1% outliers | 5.70 | 5.12 | 4.12 | 3.55 | 5.15 | 4.60 | 3.13 | 5.63 | 2.66 | 4.79 | 4.04-4.06 |
| + 0.5% outliers | 5.70 | 5.11 | 4.12 | 3.54 | 5.14 | 4.59 | 3.13 | 5.59 | 2.61 | 4.78 | 4.16-4.19 |
| + 1.0% outliers | **5.69** | **5.10** | **4.11** | **3.54** | **5.13** | **4.59** | **3.13** | **5.57** | **2.60** | **4.78** | 4.32-4.35 |
| int3 | 10.87 | 8.69 | 6.82 | 6.37 | 22.71 | 18.26 | 7.68 | 125 | 158 | 7.64 | 3.00-3.02 |
| int3-gs128 | 6.17 | 5.47 | 4.44 | 3.78 | 6.15 | 5.34 | 3.33 | 7.50 | 4.11 | 5.16 | 3.15 |
| int3-gs64 | 5.93 | 5.29 | 4.26 | 3.66 | 5.64 | 4.98 | 3.23 | 6.38 | 3.23 | 5.00 | 3.30 |
| nf3 | 7.33 | 6.21 | 5.46 | 4.44 | 9.96 | 9.50 | 4.06 | 61.07 | 98.64 | 6.30 | 3.00-3.03 |
| nf3-gs128 | 6.26 | 5.52 | 4.54 | 3.83 | 6.21 | 5.43 | 3.38 | 7.79 | 4.74 | 5.23 | 3.25 |
| nuq3 | 5.94 | 5.32 | 4.34 | 3.68 | 5.39 | 4.82 | 3.23 | 6.10 | 2.99 | 4.98 | 3.00-3.02 |
| + 0.1% outliers | 5.82 | 5.22 | 4.21 | 3.62 | 5.27 | 4.69 | 3.19 | 5.96 | 2.96 | 4.91 | 3.04-3.06 |
| + 0.5% outliers | 5.76 | 5.16 | 4.16 | 3.59 | 5.20 | 4.65 | 3.16 | 5.74 | 2.69 | 4.84 | 3.16-3.19 |
| + 1.0% outliers | **5.75** | **5.15** | **4.15** | **3.57** | **5.18** | **4.62** | **3.15** | **5.67** | **2.66** | **4.82** | 3.32-3.35 |
| int2 | 11779 | 69965 | 1470 | 7272 | 4708 | 3943 | 976 | 2841 | 2164 | 573 | 2.00-2.02 |
| int2-gs128 | 37.37 | 41.77 | 16.49 | 13.63 | 118 | 93.09 | 18.31 | 200 | 2092 | 51.96 | 2.14 |
| int2-gs64 | 11.09 | 9.84 | 6.60 | 5.54 | 25.69 | 26.83 | 5.93 | 57.82 | 43.69 | 12.47 | 2.28 |
| nf2 | 3210 | 5785 | 2044 | 5367 | 13601 | 4036 | 3680 | 30492 | 9486 | 903 | 2.00-2.03 |
| nf2-gs128 | 351 | 141 | 60.97 | 31.69 | 635 | 642 | 71.21 | 1024 | 3091 | 253 | 2.25 |
| nuq2 | 8.47 | 7.29 | 6.08 | 9.19 | 11.20 | 23.34 | 4.18 | 16.63 | 6.79 | 6.87 | 2.00-2.02 |
| + 0.1% outliers | 6.82 | 5.72 | 4.83 | 4.00 | 6.38 | 5.33 | 3.54 | 7.97 | 4.77 | 5.83 | 2.04-2.06 |
| + 0.5% outliers | 6.24 | 5.49 | 4.45 | 3.80 | 5.59 | 4.95 | 3.33 | 6.53 | 3.18 | 5.28 | 2.16-2.19 |
| + 1.0% outliers | **6.05** | **5.39** | **4.41** | **3.72** | **5.47** | **4.85** | **3.28** | **6.29** | **2.95** | **5.14** | 2.32-2.35 |

**Table 18:** *Full evaluation of our method for different models on all LLaMA, Llama-2, Llama-3, and Mistral models using the perplexity on Wikitext-2. KVQuant results are using pre-RoPE per-channel quantization for Keys. Average bit widths assume a sequence length of 128K (ignoring context length limits for the models). Note that ATOM and FlexGen use 4-bit quantization with group sizes of 128 and 64 with uniform quantization, respectively, and we extend their methods to 3-bit and 2-bit quantization. We leverage Attention-Sink Aware quantization for all bit widths. We used post-RoPE quantization for all baseline methods since it achieves higher accuracy when quantizing Keys per-token as shown in Appendix P.*

| Method | LLaMA 7B | 13B | 30B | 65B | Llama-2 7B | 13B | 70B | Llama-3 8B | 70B | Mistral-7B | Avg. Num. Bits |
|---|---|---|---|---|---|---|---|---|---|---|---|
| baseline | 5.68 | 5.09 | 4.10 | 3.53 | 5.12 | 4.57 | 3.12 | 5.54 | 2.59 | 4.76 | 16 |
| int4 | 5.98 | 5.32 | 4.34 | 3.73 | 5.66 | 5.01 | 3.31 | 7.89 | 14.03 | 4.97 | 4.00-4.02 |
| nf4 | 5.87 | 5.23 | 4.25 | 3.63 | 5.47 | 4.90 | 3.22 | 6.42 | 3.86 | 4.91 | 4.00-4.03 |
| ATOM-4bit | 5.77 | 5.16 | 4.16 | 3.57 | 5.32 | 4.71 | 3.16 | 5.78 | 2.73 | 4.82 | 4.16 |
| Flexgen-4bit | 5.73 | 5.14 | 4.14 | 3.56 | 5.25 | 4.66 | 3.14 | 5.68 | 2.66 | 4.80 | 4.31 |
| KVQuant-4bit | 5.72 | 5.13 | 4.13 | 3.55 | 5.16 | 4.60 | 3.13 | 5.60 | 2.62 | 4.80 | 4.00-4.02 |
| KVQuant-4bit-1% | **5.69** | **5.10** | **4.11** | **3.54** | **5.13** | **4.58** | **3.13** | **5.56** | **2.60** | **4.78** | 4.32-4.35 |
| int3 | 10.87 | 8.69 | 6.82 | 6.37 | 22.71 | 18.26 | 7.68 | 125 | 158 | 7.64 | 3.00-3.02 |
| nf3 | 7.33 | 6.21 | 5.46 | 4.44 | 9.96 | 9.50 | 4.06 | 61.07 | 98.64 | 6.30 | 3.00-3.03 |
| ATOM-3bit | 6.17 | 5.47 | 4.44 | 3.78 | 6.15 | 5.34 | 3.33 | 7.50 | 4.11 | 5.16 | 3.15 |
| FlexGen-3bit | 5.93 | 5.29 | 4.26 | 3.66 | 5.64 | 4.98 | 3.23 | 6.38 | 3.23 | 5.00 | 3.30 |
| KVQuant-3bit | 5.87 | 5.25 | 4.25 | 3.63 | 5.34 | 4.71 | 3.18 | 5.84 | 2.72 | 4.98 | 3.00-3.02 |
| KVQuant-3bit-1% | **5.75** | **5.14** | **4.15** | **3.57** | **5.17** | **4.61** | **3.15** | **5.64** | **2.63** | **4.82** | 3.32-3.35 |
| int2 | 11779 | 69965 | 1470 | 7272 | 4708 | 3943 | 976 | 2841 | 2164 | 573 | 2.00-2.02 |
| nf2 | 3210 | 5786 | 2044 | 5367 | 13601 | 4036 | 3680 | 30492 | 9486 | 903 | 2.00-2.03 |
| ATOM-2bit | 37.37 | 41.77 | 16.49 | 13.63 | 118 | 93.09 | 18.31 | 200 | 2092 | 51.96 | 2.14 |
| FlexGen-2bit | 11.09 | 9.84 | 6.60 | 5.54 | 25.69 | 26.83 | 5.93 | 57.82 | 43.69 | 12.47 | 2.28 |
| KVQuant-2bit | 7.23 | 5.82 | 4.87 | 4.03 | 7.03 | 9.59 | 3.45 | 7.04 | 3.49 | 6.84 | 2.00-2.02 |
| KVQuant-2bit-1% | **6.01** | **5.36** | **4.35** | **3.70** | **5.41** | **4.78** | **3.26** | **6.04** | **2.82** | **5.14** | 2.32-2.35 |

**Table 19:** *Full evaluation of our method for different models on all LLaMA, Llama-2, Llama-3, and Mistral models using the perplexity on C4. KVQuant results are using pre-RoPE per-channel quantization for Keys. Average bit widths assume a sequence length of 128K (ignoring context length limits for the models). Note that ATOM and FlexGen use 4-bit quantization with group sizes of 128 and 64 with uniform quantization, respectively, and we extend their methods to 3-bit and 2-bit quantization. We leverage Attention-Sink Aware quantization for all bit widths. We used post-RoPE quantization for all baseline methods since it achieves higher accuracy when quantizing Keys per-token as shown in Appendix P.*

| Method | LLaMA | | | | Llama-2 | | | Llama-3 | | Mistral-7B | Avg. Num. Bits |
|--------|-------|-----|-----|-----|---------|-----|-----|---------|-----|------------|----------------|
| | 7B | 13B | 30B | 65B | 7B | 13B | 70B | 8B | 70B | | |
| baseline | 7.08 | 6.61 | 5.98 | 5.62 | 6.63 | 6.05 | 4.97 | 7.10 | 5.78 | 5.71 | 16 |
| int4 | 7.40 | 6.82 | 6.18 | 5.75 | 7.31 | 6.59 | 5.12 | 8.79 | 14.36 | 5.91 | 4.00-4.02 |
| nf4 | 7.27 | 6.74 | 6.10 | 5.69 | 7.09 | 6.45 | 5.06 | 7.93 | 6.58 | 5.85 | 4.00-4.03 |
| ATOM-4bit | 7.16 | 6.67 | 6.02 | 5.65 | 6.87 | 6.20 | 5.00 | 7.32 | 6.03 | 5.76 | 4.16 |
| FlexGen-4bit | 7.12 | 6.64 | 6.00 | 5.63 | 6.79 | 6.15 | 4.99 | 7.23 | 5.84 | 5.75 | 4.31 |
| KVQuant-4bit | 7.11 | 6.64 | 6.00 | 5.63 | 6.68 | 6.08 | 4.99 | 7.18 | 5.80 | 5.75 | 4.00-4.02 |
| KVQuant-4bit-1% | **7.09** | **6.62** | **5.99** | **5.62** | **6.64** | **6.06** | **4.98** | **7.12** | **5.78** | **5.72** | 4.32-4.35 |
| int3 | 12.97 | 10.95 | 9.13 | 8.27 | 30.14 | 28.57 | 16.00 | 63.75 | 301 | 8.84 | 3.00-3.02 |
| nf3 | 8.90 | 7.84 | 7.43 | 6.37 | 14.92 | 13.75 | 5.96 | 68.38 | 148 | 7.27 | 3.00-3.03 |
| ATOM-3bit | 7.62 | 6.93 | 6.24 | 5.79 | 8.00 | 7.06 | 5.16 | 9.00 | 6.72 | 6.08 | 3.15 |
| FlexGen-3bit | 7.34 | 6.78 | 6.11 | 5.70 | 7.29 | 6.59 | 5.08 | 7.92 | 6.18 | 5.92 | 3.30 |
| KVQuant-3bit | 7.25 | 6.72 | 6.06 | 5.68 | 6.89 | 6.18 | 5.03 | 7.43 | 5.90 | 5.92 | 3.00-3.02 |
| KVQuant-3bit-1% | **7.13** | **6.64** | **6.00** | **5.63** | **6.69** | **6.09** | **4.99** | **7.19** | **5.81** | **5.76** | 3.32-3.35 |
| int2 | 10892 | 100870 | 1411 | 7216 | 4708 | 4220 | 814 | 2113 | 1977 | 477 | 2.00-2.02 |
| nf2 | 2850 | 4680 | 1617 | 5190 | 13081 | 4176 | 3217 | 78331 | 7616 | 1102 | 2.00-2.03 |
| ATOM-2bit | 43.49 | 56.25 | 21.07 | 17.05 | 113 | 97.04 | 23.67 | 135 | 3734 | 50.73 | 2.14 |
| FlexGen-2bit | 13.91 | 13.36 | 8.49 | 7.34 | 35.21 | 40.40 | 8.28 | 50.78 | 42.27 | 13.83 | 2.28 |
| KVQuant-2bit | 8.52 | 7.32 | 6.67 | 5.96 | 9.49 | 15.36 | 5.30 | 10.06 | 6.45 | 7.63 | 2.00-2.02 |
| KVQuant-2bit-1% | **7.33** | **6.78** | **6.11** | **5.70** | **6.96** | **6.25** | **5.08** | **7.60** | **5.96** | **6.05** | 2.32-2.35 |

## P    Post-RoPE Per-Token Quantization Ablation

Table 20 shows perplexity evaluation on Wikitext-2 for the LLaMA-7B model with uniform quantization, with Keys quantized pre-RoPE and post-RoPE. These results show that post-RoPE Key quantization is superior to pre-RoPE Key quantization when quantizing Keys per-token. This is because when rotating an outlier channel with large average magnitude and another channel with smaller average magnitude together, at some positions in the sequence, part of the magnitude from the outlier channel will be shifted to the smaller channel. This partially mitigates the impact of the outlier channel on skewing the quantization range for some of the tokens in the sequence. As such, for our baseline comparisons, we use post-RoPE per-token Key quantization to serve as a stronger baseline.

**Table 20:** *Model accuracy when using pre-RoPE and post-RoPE quantization for LLaMA-7B with per-token Key quantization. Our experiments demonstrate that post-RoPE quantization is superior when using per-token Key quantization. Therefore, we decided to use these results for baseline comparison with per-token quantization.*

| Datatype | Perplexity (Pre-RoPE) | Perplexity (Post-RoPE) |
|---|---|---|
| fp16 | 5.68 | 5.68 |
| int4 | 6.02 | 5.98 |
| int4-gs128 | 5.76 | 5.77 |
| int3 | 14.68 | 10.87 |
| int3-gs128 | 6.28 | 6.17 |

## Q    Experiments on Calibration Data Robustness

To evaluate the robustness of our quantization method to the choice of calibration datasets, we measure the perplexity on Wikitext-2 and C4 using different quantized models calibrated with Wikitext-2 and C4. As shown in Table 21, the resulting perplexity numbers remain similar even when the models are calibrated with out-of-domain examples (e.g., calibrated with Wikitext-2 and evaluated on C4, and vice versa), demonstrating the robustness of the calibration process to the choice of data.

**Table 21:** *Perplexity (PPL) results on Wikitext-2 and C4 using different quantization schemes, calibrated using Wikitext-2 and C4.*

| Datatype | Wikitext-2 | | C4 | |
|---|---|---|---|---|
| | Calib. with Wikitext-2 | Calib. with C4 | Calib. with Wikitext-2 | Calib. with C4 |
| 4-bit, 1% sparsity | 5.69 | 5.70 | 7.09 | 7.09 |
| 3-bit, 1% sparsity | 5.75 | 5.75 | 7.13 | 7.13 |
| 2-bit, 1% sparsity | 6.05 | 6.07 | 7.38 | 7.38 |

## R    Kernel Implementation Details

We implemented 4-bit lookup table-based kernels for matrix-vector multiplication between the Key or Value activations (packed as a lookup table (LUT) plus indices into the LUT per-element) and a full-precision activation vector. These kernels load the compressed Key and Value activations and dequantize them only as needed in order to minimize memory bandwidth utilization. All arithmetic is performed in fp16. The lookup table entries are the values of the sensitivity-weighted non-uniform datatype for that particular layer scaled according to the range of activations that need to be represented [8].

When selecting between the Compressed-Sparse Column (CSC format) and the Compressed-Sparse Row (CSR) format for storing the outliers for the Keys and Values, we needed to consider how easy it would be to append new vectors. When using CSC format for the Key matrix, we only need to append a single element to the column vector, as well as one new element to the row and value vectors per nonzero element in that new column. If we used CSR format, we would need to insert the new column and value elements in the middle of the existing column and value vectors, and we would

need to recompute the elements of the row vector. When using CSR format for the Value matrix, we only need to append a single element to the row vector, as well as one new element to the column and value vectors per nonzero element in that new row. If we used CSC format, we would need to insert the new row and value elements in the middle of the existing row and value vectors, and we would need to recompute the elements of the column vector. We therefore used the CSC format for the Key matrices and the CSR format for the Value matrices.

One challenge with efficiently processing the sparse matrix-dense vector operation is that the sparsity distribution may be unbalanced. This poses a challenge for efficiently processing the sparse matrix on a GPU as there can be different numbers of nonzeros to process per thread. We therefore leverage a *balanced* sparse matrix-dense vector kernel based on [10, 17], which assigns an equal number of nonzeros per thread. This has greater synchronization overhead than assigning a single thread for an entire row or column when processing CSR/CSC matrices, but it leads to a more balanced work assignment between threads. We set the number of threads such that there were 10 nonzero values assigned to each thread. The dense non-uniform kernel and balanced sparse kernels are launched in one call to avoid overhead from summing the output vectors from these separate operations.

A second potential challenge is that the quantization dimension for the Keys and Values is not aligned with the reduction dimension for the respective matrix-vector operations. If we used per-channel or per-token lookup tables for dequantization, we would need to load from a different lookup table for each element as we iterate along the reduction dimension. However, since our implementation uses a single per-layer datatype that is rescaled per-vector, we can load a single scaling factor and zero-point for each channel / token and broadcast it across all threads (while doing a table lookup from the shared per-layer datatype, which is duplicated such that each thread can access it efficiently in parallel). The use of a single per-layer datatype that is rescaled per-vector therefore allows for much more efficient GPU kernel implementations in the context of per-channel Key / per-token Value quantization.

Table 22 shows a detailed breakdown of kernel runtime, including how much time is spent packing vectors into the compressed format and how much time is spent on the dense and sparse matrix-vector multiplications. For the fp16 baselines, we benchmarked runtime by averaging across 1000 iterations. When benchmarking our dense-and-sparse kernels (both for packing and matrix-vector multiplication), since the runtime of the sparse kernels can vary based on the sparsity pattern, we ran the LLaMA-2-7B-32K model and collected activations at each layer, and we then measured the average runtime across 1000 iterations for each layer in the model. We find that even with 1% sparsity, we can attain significant speedups of up to $1.7\times$ relative to the fp16 matrix-vector multiply kernels, demonstrating how our methodology facilitates efficient inference with a low-precision quantized KV cache.

**Table 22:** *Average latency (in microseconds) for the Key and Value dense nuq4 kernels as well as for the sparse kernels (with 1% outliers), benchmarked on an A6000 GPU for the LLaMA-2-7B-32K model. Benchmarking results are reported for different sequence lengths (l). fp16 matrix-vector multiplication latencies are included for reference, and the Key multiplication time also includes the time to apply RoPE to the newly appended Key vector. We find that our dense-and-sparse approach (even with 1% outliers) provides latency benefits relative to the fp16 baseline, even when accounting for the time to compress activations online.*

| Activation | Operation | l=2K | l=4K | l=16K |
|---|---|---|---|---|
| Key | fp16 Matvec | 33.3 | 59.1 | 219.4 |
| Key (nuq4-1%) | Packing | 4.5 | 4.5 | 4.5 |
| | Dense Matvec | 13.0 | 24.1 | 87.6 |
| | Sparse Matvec | 8.1 | 11.2 | 34.2 |
| | Total Latency | 25.6 | 39.9 | 126.3 |
| Value | fp16 Matvec | 26.0 | 50.2 | 203.7 |
| Value (nuq4-1%) | Packing | 4.1 | 4.1 | 4.1 |
| | Dense Matvec | 10.0 | 17.8 | 62.0 |
| | Sparse Matvec | 7.9 | 15.9 | 58.2 |
| | Total Latency | 22.1 | 37.9 | 124.5 |

# S   Limitations

While our work enables accurate long-context length inference by reducing the memory requirements, there is significant work required for training long context length models with greater than 100K context length. This work is orthogonal to our efforts, which are constrained to efficient inference with long context length models. Additionally, our latency benchmarking results currently focus on memory-bandwidth bound generation rather than prompt processing (where we need to compress multiple Keys and Values at once). Finally, in the current end-to-end implementation, there are inefficiencies in how memory allocation is handled for updating the sparse matrix (where the data corresponding to the previous tokens have to be copied when concatenating them with the data from the new token). In future work, we plan to optimize this by doing blocked allocation to avoid overheads from reallocating memory.

