# OpenReview forum: "KVQuant: Towards 10 Million Context Length LLM Inference with KV Cache Quantization"
_NeurIPS.cc/2024/Conference — NeurIPS 2024 poster_

### Official Review · Reviewer_TbGM · 2024-07-10

**Soundness:** 4
**Presentation:** 4
**Contribution:** 3
**Rating:** 7
**Confidence:** 4

**Summary:**

KVQuant presents a method for applying low-bit activation quantization to the Key-Value Cache, a major bottleneck in long context LLM's generation inference. The authors propose strategies (per-channel, per-token, pre-RoPE) tailored to the distribution characteristics of keys and values, as well as the distribution shift introduced by RoPE. They effectively mitigate the overhead introduced by their approach through kernel optimization.

For the quantization process, they draw inspiration from SqueezeLLM, employing a non-uniform, dense-and-sparse method to handle outliers efficiently. They also use offline calibration to manage dynamically induced quantization scale and outliers from KV activations.

The proposed KVQuant method demonstrates the least perplexity degradation in low-bit kv-cache quantization compared to FP16. Additionally, it outperforms the latest KV Quantization baseline (KIVI) based on group-wise quantization in terms of retrieval accuracy and LongBench performance.

**Strengths:**

- Paper's writing is clear and easy to follow. It clearly explains why the kv-cache is the main bottleneck in long context scenario and highlights the need for quantization in memory-bound situations during LLM generation inference, making the importance of the paper easily understandable for the reader.
- Beyond merely presenting performance, the paper minimizes the overhead of the proposed methods (Per-Channel Key Quantization, applying RoPE on-the-fly, offline calibration) through kernel optimization.
- The strengths of this paper are clearly highlighted through comparisons with recent KV quantization baseline, KIVI.
- The paper shows evaluation results not only with generation style (PPL) but also in long context prefill processing results (longbench, longeval).

**Weaknesses:**

Since this paper is very solid, I don't see any major weakness in this paper. However, it is more focused on technical strategies for effectively applying quantization to the KV cache rather than presenting innovative or novel ideas. However, such efforts are crucial for making LLMs more accessible, so it is not considered a major weakness of the paper.

As a minor weakness, due to the extensive experimental results and content, there are many appendices linked in the main text and frequent parenthetical explanations, which can make the paper somewhat challenging to follow. There is room to improve readability by filtering out and emphasizing key points in the main text.

**Questions:**

- Do you have any insights into the cause of the disparate characteristics in the distributions of keys and values?
- Can the distribution characteristics of Key and Value be generalized to widely used popularized LLMs? (e.g., Mistral, Gemma-2, Phi-3, …)
- I am curious about the specific method used to measure PPL. I might have missed it, but when measuring generation PPL on wikitext and C4, did the authors use a teacher forcing style by providing all input tokens at once, applying quantization to the KV embeddings, and measuring loss with the final logits for all tokens? It would be helpful to have a detailed explanation of how PPL was measured.

**Limitations:**

Limitation section in Appendix

---

> ### Author Rebuttal · Authors · 2024-08-07
>
> Thank you for taking the time to review our paper!
>
>
> > As a minor weakness, due to the extensive experimental results and content, there are many appendices linked in the main text and frequent parenthetical explanations, which can make the paper somewhat challenging to follow. There is room to improve readability by filtering out and emphasizing key points in the main text.
>
> We appreciate the feedback on how to better structure our paper in order to represent the key contributions and takeaways from our analysis and results. In the final version, we will streamline the paper by focusing on key points in the main text (and ensuring that these are fully explained within the main text), while making sure the supplemental information is not necessary for understanding the core contributions of our work.
>
>
> > Can the distribution characteristics of Key and Value be generalized to widely used popularized LLMs? (e.g., Mistral, Gemma-2, Phi-3, …)
>
> We have observed that these characteristics are consistent across different models in our work (eg. LLaMA, Llama-2/3, Mistral). Additionally, in the rebuttal PDF we have attached the analysis of the KV cache distributions for the Gemma-2-2B and Phi-3-Mini-128K-Instruct models for a randomly selected layer. These distributions are similar to what we observed in the analysis for LLaMA-7B included in Figure 2 of our paper (i.e. both models have outlier input channels with consistent average magnitudes pre-RoPE), indicating that pre-RoPE per-channel quantization would generalize to these networks.
>
> > Do you have any insights into the cause of the disparate characteristics in the distributions of keys and values?
>
> With respect to the disparate distributions of keys and values, keys have specific outlier channels with significantly larger average magnitude than other channels, as highlighted in Figure 2. For values, we do not observe particular per-token or per-channel outliers. Given the consistent patterns that we observe across different model families and model sizes, we think that it is an interesting direction for further research to understand the root cause of the distributional characteristics of keys and values.
>
>
> > I am curious about the specific method used to measure PPL. I might have missed it, but when measuring generation PPL on wikitext and C4, did the authors use a teacher forcing style by providing all input tokens at once, applying quantization to the KV embeddings, and measuring loss with the final logits for all tokens? It would be helpful to have a detailed explanation of how PPL was measured.
>
> This is a great question! We are indeed using teacher forcing by providing all input tokens in parallel and quantizing the Key and Value embeddings on-the-fly in the forward pass. Then we measure the perplexity using the output logits of *all input tokens*. We will clarify this point in the final version of the paper.

---

> > ### Comment · Reviewer_TbGM · 2024-08-09
> >
> > Thank you for the detailed response. Hope that the revised version of the paper will better emphasize the core content in the main part. Additionally, it would be beneficial if the experimental section could further highlight the effectiveness of KVQuant in long context scenarios where KV cache compression becomes particularly important (e.g., retrieval, longbench, RAG). That said, I find this paper to be solid and insightful, and I will maintain my recommendation to accept.

---

### Official Review · Reviewer_gDca · 2024-07-11

**Soundness:** 3
**Presentation:** 3
**Contribution:** 2
**Rating:** 6
**Confidence:** 4

**Summary:**

This paper proposes KVQuant, which consists of 4 techniques for improving the performance of low-bit KV cache quantization. By observing the distribution of Key and Value cache, it proposes to use channel-wise quantization for Key cache before RoPE and token-wise quantization for Value cache. It also adopts non-uniform quantization instead of uniform quantization to improve performance. It also leverages per-vector dense-and-sparse quantization and attention sink-aware quantization to isolate outliers. Experiments on Wikitext-2 and C4 are conducted to evaluate the proposed method.

**Strengths:**

1.	The analysis of the KV cache distributions provide some insights to KV cache quantization.
2.	Based on the observation, the proposed techniques are reasonable for KV cache quantization.
3.	Latency of the implemented kernels is reported.

**Weaknesses:**

1.	The paper seems to be a combination of several existing quantization methods (or minor revision of existing methods), e.g. choosing to use per-channel quantization for Key cache, using non-uniform quantization, and dense-and-sparse quantization to improve performance. It is more like a technical report.
2.	Based on the previous point, the paper writing is not clear enough. The paper proposes too many techniques and details are not fully presented in the main text of the paper. For instance, how is the non-uniform quantization used in this paper specifically implemented?
3.	The evaluation section lacks detailed comparisons of the existing KV cache compression methods (like H2O, GEAR, KIVI).
4.	The experiments are only conducted on Wikitext-2 and C4 datasets and reported the PPL. The effectiveness is not very well proven. Other challenging tasks such as reading comprehension, math problem solving, or code generation should be evaluated.

**Questions:**

1.	Quantization signposts appear multiple times in the article, what exactly is this?
2.	The paper uses a calibration set to optimize the quantization parameters. Will this lead to overfitting of the model? Will calibrating a quantized model on one dataset affect its performance on other datasets?

---

> ### Author Rebuttal · Authors · 2024-08-07
>
> Thank you for the detailed feedback on our work!
>
> >The paper seems to be a combination of several existing quantization methods), e.g. choosing to use per-channel quantization for Key cache, using non-uniform quantization, and dense-and-sparse quantization to improve performance.
>
> With respect to the novelty of our approach, we would like to highlight multiple contributions from our work:
>
> - To the best of our knowledge, our work was the first to observe the impacts of RoPE on per-channel outliers in Keys and to propose *pre-RoPE per-channel key quantization* to mitigate this challenge.
> - Our method leverages non-uniform layer-wise datatypes derived from calibration data, which facilitates more accurate quantization online for activations than prior works using fixed datatypes, without compromising on efficiency.
> - Our work shows that dense-and-sparse quantization can be applied efficiently for activations, and demonstrates the necessity of per-vector outlier detection for applying dense-and-sparse quantization to activations with outlier channels (which was not explored in prior works on weight quantization).
>
> >The paper writing is not clear enough. The paper proposes too many techniques and details are not fully presented in the main text of the paper.
>
> We appreciate the feedback. In the revised version, we will ensure to highlight all key takeaways in the main text such that they can be understood without referencing the appendix.
>
> >The experiments are only conducted on Wikitext-2 and C4 datasets and reported the PPL. Other challenging tasks such as reading comprehension, math problem solving, or code generation should be evaluated.
>
> In our submission, we indeed included benchmarks other than PPL evaluation such as **passkey retrieval** results (Section 4.2, Table 2), as well as **LongBench** evaluation (Appendix Q, Table 18), which includes a range of downstream tasks such as QA, summarization, code generation, and in-context learning.
>
> Below, as a part of our rebuttal, we include additional evaluation of KVQuant on various benchmarks including **XSum** (summarization), **GSM8K** (math reasoning), and **RULER** (retrieval, QA, aggregation, multi-hop tracing). Please check our response below for more details.
>
> >The evaluation section lacks detailed comparisons of the existing KV cache compression methods (like H2O, GEAR, KIVI).
>
> We appreciate the suggestions for relevant comparisons for our work.
>
> ### 1. Comparison against KIVI
>
> In our paper, we included comparisons with KIVI for passkey retrieval as well as LongBench evaluation with LLaMA-2-7B-32K. On passkey retrieval (Table 2), KIVI attains 68-76% with an average bit-width of 3, whereas KVQuant achieves an accuracy of >98% even with 2-bit quantization. On LongBench (Table 18), 3-bit KVQuant shows only 0.75 point degradation from the FP16 baseline, which is much smaller than the 1.92 point degradation of KIVI with a similar average bit-width (31.96 → 31.21 vs 30.04).
>
> In the table below, we additionally compare KVQuant and KIVI on the RULER benchmark suite using LLaMA-2-7B-32K. Here, **3-bit KVQuant achieves ~14% better score against KIVI with similar average bit-width**.  Furthermore, our **2-bit KVQuant achieves similar accuracy to KIVI with 1.4x smaller bit-width**. The results on passkey retrieval, LongBench, and RULER demonstrate how our method provides improved accuracy on long context-length tasks compared to KIVI.
>
> **LLaMA-2-7B-32K on RULER**
>
> |**Config**|**Avg Bit Width**|**Avg Accuracy**|
> |-|-|-|
> |fp16 Baseline|16|56.40|
> |KIVI (2-bit, group size 32, residual size 128)|3.05|39.78|
> |KVQuant (3-bit, 1% sparsity)|3.33|**53.65**|
> |KVQuant (2-bit, 1% sparsity)|2.33|36.54|
>
> * Accuracies for each subtask are in Table 1 of the rebuttal PDF
>
> ### 2. Comparison against H2O and GEAR
>
> We have also included additional comparisons with H2O and GEAR to provide further insight into how our approach compares with other KV cache compression methods. The table below provides a comparison with H2O on XSum with 5-shot prompting for LLaMA-7B, using the evaluation script from the H2O repo.  As demonstrated in the tables, our method achieves **consistently better ROUGE scores (up to ~3 point)** compared to H2O with the same compression rates. Furthermore, our method does not show performance degradation up to 5x compression rate.
>
> **LLaMA-2-7B on XSUM**
>
> |**Config**|**R-1**|**R-2**|**R-L**|
> |-|-|-|-|
> |fp16 Baseline|31.2|11.9|26.2|
> |H2O (4x compression)|30.3|10.9|25.2|
> |KVQuant (4-bit, 1% sparsity, 3.7x compression)|**31.3**|**12.0**|**26.4**|
> |H2O (5x compression)|29.6|10.4|24.4|
> |KVQuant (3-bit, 1% sparsity, 4.8x compression)|**31.2**|**11.6**|**26.1**|
> |H2O (7x compression)|26.2|8.6|21.8|
> |KVQuant (2-bit, 1% sparsity, 6.8x compression)|**29.3**|**10.4**|**24.4**|
>
> Additionally, in the table below, we also included GSM8K comparisons with both GEAR and H2O using 8-shot CoT prompting using the environment from the GEAR repo. Here, our method achieves **better GSM8K accuracy compared to both H2O and GEAR even with a bigger compression rate**.
>
> **LLaMA-2-7B on GSM8K**
>
> |**Configuration**|**Accuracy**|
> |-|-|
> |fp16 Baseline|16.30|
> |H2O (2x compression)|6.82|
> |GEAR (3x compression)|14.17|
> |KVQuant (4-bit, 1% sparsity, 3.7x compression)|**14.71**|
>
> >What exactly is the word quantization signpost?
>
> Quantization signposts refers to the discrete representable values when quantizing to reduced precision. For example, if the representable values with 2-bit quantization are [-0.7, -0.2, 0.2, 0.7], then these 4 values are the signposts.
>
> >Will calibration set lead to overfitting of the model? Will calibrating a quantized model on one dataset affect its performance on other datasets?
>
> Please refer to our response to Reviewer 2 for our ablation study which demonstrates that our method is not overfitting to the calibration dataset. Furthermore, all experiments in our paper were calibrated using 16 examples from WikiText2 and evaluated on different downstream tasks.

---

> > ### Comment · Reviewer_gDca · 2024-08-09
> > **Thank you for your response.**
> >
> > Most of my concerns are addressed. So I'm willing to raise the score.

---

### Official Review · Reviewer_j7z6 · 2024-07-13

**Soundness:** 4
**Presentation:** 3
**Contribution:** 3
**Rating:** 7
**Confidence:** 2

**Summary:**

This paper proposes KVQuant, a quantization framework for enabling long context window inference through compressing KV cache activations. Specifically, the KVQuant framework incorporates several techniques including per-channel key quantification, per-RoPE key quantification, non-uniform KV cache quantification, and per-vector dense-and-sparse quantification. Experiments on several datasets and LLMs show that KVQuant can achieve low precision (3-bit) quantization with a relatively low impact on model performance.

**Strengths:**

- Addressing memory constraints on context window length by compressing KV cache activations is a timely and important research area.

- Handling outliers before RePE and non-uniform quantization by considering sensitivity are good insights.

- Extensive experiments were conducted, covering prominent models such as  Llama, Llama-2, Llama-3, and Mistral, on both Wikitext-2 and C4, which verified that KVQuant can achieve 3-bit quantization with < 0.1 perplexity degradation.

- Custom CUDA kernels are developed to accelerate inference.

**Weaknesses:**

In the per-vector dense-and-sparse quantization step, the numerical outliers are stored as high precision in a separate sparse matrix. It is not quite clear how to achieve this in practice in a hardware-friendly way without affecting inference speed. Additionally, the offline calibration step makes the implicit assumption on the access to data samples that are from a similar distribution as data from inference time.

**Questions:**

Please refer to weaknesses.

**Limitations:**

The authors discussed the limitations in Appendix T.

---

> ### Author Rebuttal · Authors · 2024-08-07
>
> Thank you for taking the time to review our paper.
>
>
> > In the per-vector dense-and-sparse quantization step, the numerical outliers are stored as high precision in a separate sparse matrix. It is not quite clear how to achieve this in practice in a hardware-friendly way without affecting inference speed.
>
> It is correct that storing the numerical outliers separately in a sparse high-precision matrix can add considerable inference latency if the sparsity percentage is too high. However, we have observed that KV cache activations have a relatively small percentage of outlier elements (<1%) which need to be stored separately to alleviate quantization difficulty. By storing only this small percentage of outliers in a separate sparse matrix, we observe significant accuracy benefits; as highlighted in Figure 1, isolating 1% outliers improves the perplexity on Wikitext-2 for the LLaMA-7B model from 5.94 to 5.75, achieving <0.1 PPL degradation with 3-bit quantization. Since the sparsity percentage is extremely low, we find that the sparse matrix / dense vector computation doesn’t add significant runtime overhead, as shown in Appendix S. Our approach can therefore store outliers separately in higher precision to maintain high accuracy with low memory consumption, without adding significant latency overhead.
>
>
> >  Additionally, the offline calibration step makes the implicit assumption on the access to data samples that are from a similar distribution as data from inference time.
>
> In our experiments, we have only used a small amount of data from the Wikitext-2 training set for calibration (16 samples of length 2048), and then we do not recalibrate when running inference on different downstream tasks. We observe strong performance across a range of downstream tasks (such as LongBench in Table 18 and passkey retrieval in Table 2), demonstrating that the calibration process is not sensitive to the choice of calibration data.
>
> To further support this claim, we have included an ablation in the table below showing the perplexity of LLaMA-7B on Wikitext-2 and C4 when we use each of the two datasets as our calibration dataset. These results demonstrate that the accuracy is relatively unchanged when running calibration using different training sets.
>
> | **Datatype**       | **Wikitext-2 PPL, Calibrated using Wikitext-2** | **Wikitext-2 PPL, Calibrated using C4** | **C4 PPL, Calibrated using Wikitext-2** | **C4 PPL, Calibrated using C4** |
> | ------------------ | ----------------------------------------------- | --------------------------------------- | --------------------------------------- | ------------------------------- |
> | 4-bit, 1% sparsity | 5.69                                            | 5.70                                    | 7.09                                    | 7.09                            |
> | 3-bit, 1% sparsity | 5.75                                            | 5.75                                    | 7.13                                    | 7.13                            |
> | 2-bit, 1% sparsity | 6.05                                            | 6.07                                    | 7.38                                    | 7.38                            |

---

> > ### Comment · Reviewer_j7z6 · 2024-08-10
> >
> > Thanks for the rebuttal and the additional results. I have no further comments and will keep my score in support of this paper.

---

### Official Review · Reviewer_MBTK · 2024-07-14

**Soundness:** 3
**Presentation:** 3
**Contribution:** 3
**Rating:** 5
**Confidence:** 3

**Summary:**

The paper presents KVQuant, a low precision quantization method for KV cache activations in LLMs to reduce memory consumption during inference with large context windows. KVQuant applies per-channel pre-RoPE quantization on Key cache, exploits non-uniform datatype for quantization, and keeps the per-vector outliers in full precision. KVQuant achieves neligible perplexity degradation with 3-bit quantization on Wikitext-2 and C4, outperforming existing methods, and allows serving LLaMA-7B with up to 1 million context length on a single A100-80GB GPU.

**Strengths:**

+ The paper addresses a crucial issue in the use of LLMs for long-context applications, where memory capacity is a significant limitation.
+ The paper is well-written and easy to follow.
+ The observation regarding the distribution of Key activations and its impact on quantization is particularly insightful, contributing valuable knowledge to KV cache quantization.
+ The paper includes practical, real-world measurements of speedup using specialized CUDA kernels, demonstrating the tangible benefits of their approach.
+ A thorough ablation study is presented, which helps in understanding the design choices made in KVQuant.

**Weaknesses:**

- The paper does not sufficiently clarify the baseline methods and implementations used for comparison.

**Questions:**

- What is the FP16 baseline implementation when evaluating speedup of the KVQuant kernel?
- Compared to Post-RoPE quantization where post-RoPE Keys are stored, why does recalculating the RoPE in the KVQuant implementation can achieve faster speeds?

**Limitations:**

The authors have adequately addressed the limitations.

---

> ### Author Rebuttal · Authors · 2024-08-07
>
> Thank you for taking the time to review our paper.
>
>
> > The paper does not sufficiently clarify the baseline methods and implementations used for comparison.
>
> The baseline methods that we used to compare with KVQuant are (i) uniform quantization with and without grouping, (ii) non-uniform quantization (with and without grouping) using the fixed NormalFloat datatype introduced in QLora, and (ii) KIVI, a state-of-the-art KV cache quantization method. Particularly for uniform quantization, we followed the same configurations as FlexGen and Atom, which use group sizes of 64 and 128, respectively, and we used per-token quantization for both keys and values. We quantize keys post-RoPE for all baseline approaches since it has higher accuracy for per-token key quantization (as outlined in Appendix P). Please let us know if there are any other details about our baseline approaches which require clarification.
>
> > What is the FP16 baseline implementation when evaluating speedup of the KVQuant kernel?
>
> The FP16 baseline implementations that we used to evaluate latency are the default kernels called by PyTorch for batched matmul operations. In particular, in Table 3 in the paper, the Key kernel baseline and the Value kernel baseline measure the latency for Query x Key and Attention Score x Value matmul operations, respectively. For the Key kernel, we additionally included the time for applying RoPE for a single key vector since the new key vector at each iteration needs to have RoPE applied before computing Query x Key. We will make this point clear in the revised version of our paper. Kernel latency is collected using the Pytorch CUDA profiling tool.
>
> > Compared to Post-RoPE quantization where post-RoPE Keys are stored, why does recalculating the RoPE in the KVQuant implementation can achieve faster speeds?
>
> Recalculating RoPE in the KVQuant implementation is introduced to enhance the quantization performance, rather than to achieve faster speeds and efficiency. As outlined in Section 3.2 of our paper, pre-RoPE key quantization is *necessary* for accurate key quantization, improving the perplexity on Wikitext-2 for the LLaMA-7B model by 0.82 with 3-bit quantization. Our experiments demonstrate that we can quickly recompute RoPE on-the-fly, thereby attaining the significant accuracy benefits of pre-RoPE key quantization without compromising much efficiency.

---

### Author Rebuttal · Authors · 2024-08-07

Thank you to all of the reviewers for taking the time to review our paper and to provide us with valuable feedback. We included responses for each of the questions that the reviewers have posed.

---

### Decision · Program_Chairs · 2024-09-25

**Decision:**

Accept (poster)

**Comment:**

The paper looks into the problem of compressing KV cache for more memory-efficient LLM inference. To reduce the KV cache size while retaining high accuracy, the authors introduce several techniques, including per-channel and pre-RoPE key state quantization, mixture of quantization and sparse outliers, non-uniform quantization, attention-sink aware quantization, offline calibration, and provide kernel optimizations. The result is a heavily compressed KV cache scheme while retaining model accuracy. The authors also evaluate their approach on a wide range of tasks, including long context tasks. The comparison with the current work KiVi is also a plus.

Overall, as other reviewers said, this is a solid paper that explores different quantization schemes to help reduce the KV cache size for LLMs. More interestingly, these schemes seem to compose well. Finally, the paper serves as a good reference list in terms of important factors that help reduce the KV cache size.